https://doi.org/10.1038/s42003-021-01803-0　　**OPEN**
# smiFISH and embryo segmentation for single-cell multi-gene RNA quantification in arthropods

Llilians Calvo [1,2], Matthew Ronshaugen [1] & Tom Pettini [1,2 ✉]

Recently, advances in fluorescent in-situ hybridization techniques and in imaging technology have enabled visualization and counting of individual RNA molecules in single cells. This has greatly enhanced the resolution in our understanding of transcriptional processes. Here, we adapt a recently published smiFISH protocol (single-molecule inexpensive fluorescent in-situ hybridization) to whole embryos across a range of arthropod model species, and also to non-embryonic tissues. Using multiple fluorophores with distinct spectra and white light laser confocal imaging, we simultaneously detect and separate single RNAs from up to eight different genes in a whole embryo. We also combine smiFISH with cell membrane immunofluorescence, and present an imaging and analysis pipeline for 3D cell segmentation and single-cell RNA counting in whole blastoderm embryos. Finally, using whole embryo single-cell RNA count data, we propose two alternative single-cell variability measures to the commonly used Fano factor, and compare the capacity of these three measures to address different aspects of single-cell expression variability.

[1] Faculty of Biology, Medicine and Health, The University of Manchester, Manchester M13 9PT, UK. [2] These authors contributed equally: Llilians Calvo, Tom Pettini. ✉email: pettini.tom@gmail.com

For many years, RNA in situ hybridization (ISH) and immunostaining have been the methods of choice for studying gene expression patterns, but have not generally been used to quantify expression levels beyond qualitative differences. This is because signal amplification steps introduce intensity variation and nonlinearity in detection precluding quantitative comparison, and off-target probe or antibody binding can produce substantial false positives[1]. Instead, quantification of gene expression has largely relied on quantitative PCR, microarrays, nanostring technology and bulk RNA-seq. These techniques usually provide only relative expression levels rather than actual RNA numbers, across a pool of cells, so a wealth of information concerning cell-to-cell variability is lost[2–6]. More recently, single-cell versions of these techniques have been developed, allowing for the first-time quantitation of cell differences in gene expression[7,8], known to be critical in influencing single-cell behaviours[9,10], differentiation[11] and disease[12]. However, the spatial context of the cells with respect to both their neighbouring cells, and to the larger tissue or embryo is often still lost[13].

Recently, these limitations have been overcome by the development of single-molecule fluorescent in situ hybridization (smFISH), which employs multiple short ~20 nt gene-specific DNA probes directly labelled with fluorophores[14,15]. When multiple short probes bind to target RNA, the single RNA molecules can be visualized and counted as discrete fluorescent spots. Accurate quantification is possible because both false positives and negatives are minimized, since a single off-target smFISH probe is below detection limits, and a false negative is unlikely as this would require that most of the ~40 probes miss the same target molecule. Furthermore, cells remain fixed within the sample rather than being dissociated, so RNA number can be quantified on a cell-by-cell basis in the spatial and temporal context of the sample. A variant of smFISH was recently developed, in which the gene-specific probes have an additional 28 nt flap sequence added to the 5′ end, rather than being directly tagged with fluorophore[16]. This flap sequence is identical for all probes in the set. The complementary sequence to the 28 nt flap is synthesized with a fluorophore of choice attached to 5′ and 3′ ends, and then prior to use, the complementary flaps are annealed, creating gene-specific probes that are now fluorophore labelled. This simple change in probe preparation vastly decreases cost, since only a single flap sequence is labelled with fluorophore, rather than each unique gene-specific sequence. Accordingly, this approach is termed single-molecule inexpensive FISH (smiFISH).

The original smiFISH publication tests the technique in cultured mammalian cells[16]. In this study, we modify the protocol, and show it to be effective in early and late embryos from five extant and emerging arthropod model species, and also in non-embryonic tissues, specifically *Drosophila* imaginal discs and ovaries. We also test the compatibility of a suite of different commercially available fluorophores, in combination with confocal imaging and a white light laser, to attain the maximum number of different RNAs that can be visualized simultaneously in the same sample. We combine smiFISH with immunofluorescence for the detection of cell membranes, and present a clearly defined analysis pipeline for whole-embryo cell segmentation in 3D image stacks, and single-cell RNA quantification for multiple genes. To enable analysis of single-cell variability, we develop an automated method for identifying the immediate neighbours of each cell in the embryo. The Fano factor, (variance/mean) is commonly used to measure cell variability in expression level[17], however, due to its limitations, here we offer two alternative measures of variability that better capture individual cell behaviour, and compare the capacity of each method to address different biological questions.

## Results

**Adaptation of smiFISH to arthropod embryos and tissues.** smiFISH was originally tested in cultured mammalian cells[16]. Here we applied the smiFISH protocol, with modifications, to embryos of five different arthropod model species—*Drosophila melanogaster* and *Drosophila virilis* (fruit flies), *Nasonia vitripennis* (parasitoid wasp), *Tribolium castaneum* (flour beetle), and *Parhyale hawaiensis* (amphipod crustacean). The evolutionary divergence times of these species is shown in Supplementary Fig. 1. We also tested *Drosophila* imaginal discs and ovaries. Our protocol simplifies the original smiFISH buffers, omitting *Escherichia coli* tRNA, BSA and vanadylribonucleoside complex. 1X PBS is swapped for 1X PBT to avoid embryo or tissue clumping, and we also increase the number and duration of washes, to account both for the fact that embryos and tissues are thicker and more complex than cells, and that complete removal of solutions between washes is less feasible.

An identical protocol was used for all species and tissues, the only minor differences were in the sample fixation method, and the final mounting (detailed in online Methods). Across species, we stained for the same two genes, *even-skipped* (*eve*) in early embryos, and *engrailed* (*en*) in later embryos (Fig. 1). Single mRNA resolution was achieved in the embryos of all species, with very low non-specific background, evident from the regions outside of stripes that are devoid of signal. In both *Drosophila* species (diverged ~50 million years ago), *eve* is expressed in seven stripes. Classically, *eve* stripes detected with normal ISH or immunostaining tend to have a discrete appearance[18–20], but here magnified panels showing the regions in between stripes at single molecule resolution reveal that *eve* is expressed throughout the entire region enclosed by the seven stripes. The stripes represent waves of alternating high and low expression. In accordance with previous observations, *eve* shows different patterns in *Tribolium*, *Parhyale* and *Nasonia*, which may reflect distinct upstream regulatory inputs, and the differing modes of segmentation in these species compared with *Drosophila*[21–23].

Imaginal discs were stained for *en* and *wingless* (*wg*) (Fig. 1). Both genes have regions within the wing disc with markedly different expression levels (*en*, magnified panel), and both sharp and diffuse boundaries (*wg*, magnified panels), presumably arising from regional differences in transcriptional regulation. Ovaries were stained for *bicoid* (*bcd*) and *nanos* (*nos*) RNAs (Fig. 1). In the stage 10 egg chamber shown, both genes are highly expressed in the nurse cells. As expected, *bcd* RNAs accumulate at high density at the anterior edge of the oocyte[24,25], with a gradient of decreasing concentration towards the posterior. *nos* RNAs are also abundant at the anterior edge of the oocyte, but additionally show the beginnings of some accumulation at the posterior pole, visible in the magnified panel[26]. The high sensitivity of smiFISH reveals the earliest onset of posterior *nos* RNA localization at stage 10, earlier than previously observed by classic in situ[26,27] (stage 12), but similar to reports from live imaging[28] and FISH staining[29] (stage 10). The majority of posterior localization of *nos* RNAs occurs during the last stages of oogenesis (stages 13/14)[26,28], so accordingly, posterior localization of RNAs in the stage 10 egg chamber shown is minimal.

**Simultaneous multi-gene visualization at single molecule resolution.** Tsanov et al. show that since smiFISH flaps are first annealed in vitro, probes using the same flap sequence but with different fluorophores can be used together without crossover[16]. Using only the X flap sequence for all smiFISH probe sets, we tested the performance of multiple fluorophores, alone and in combination, with the aim of identifying a maximum set with separable spectra, which would allow simultaneous detection of

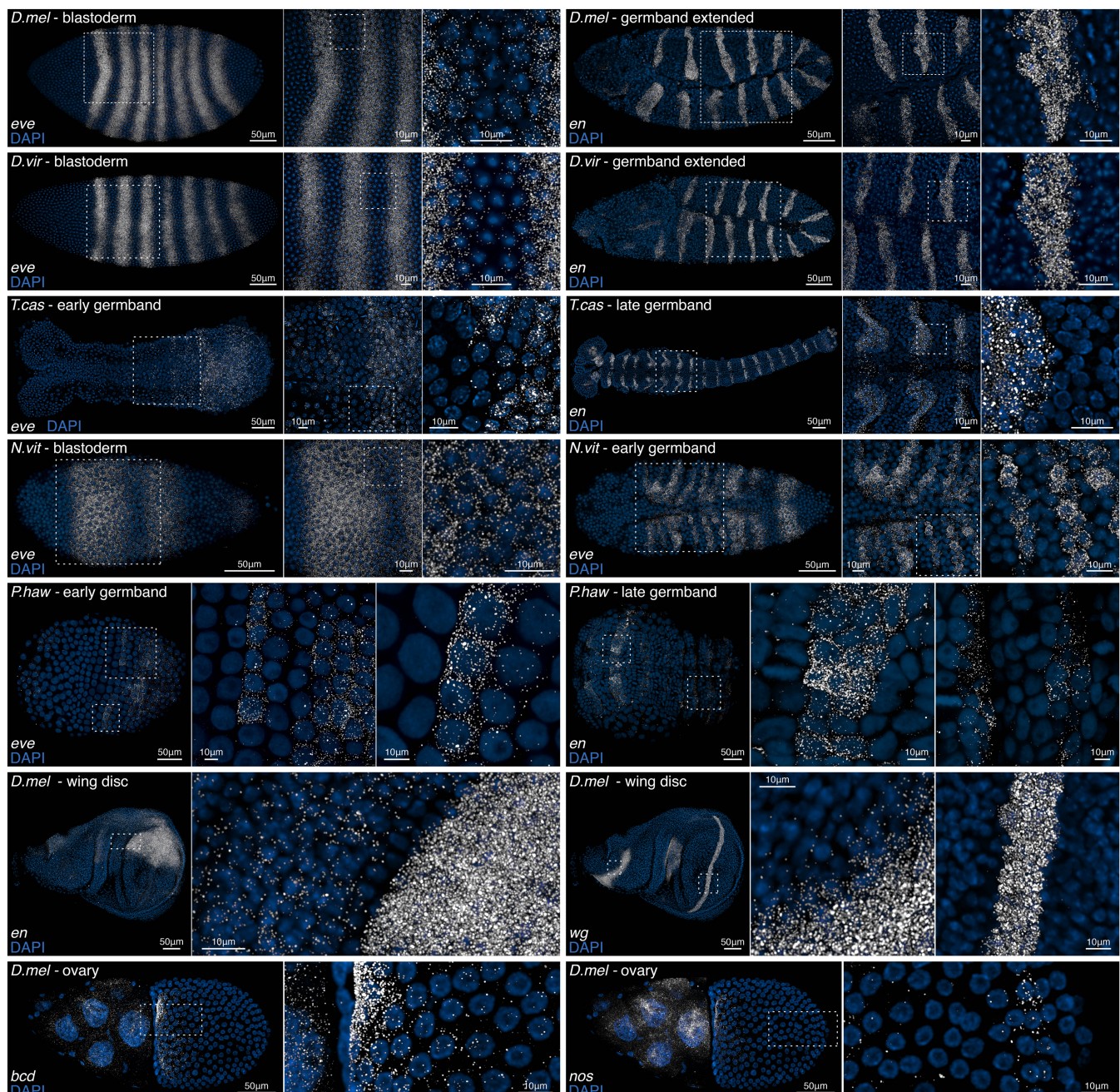

**Fig. 1 smiFISH in different arthropod species and tissues.** smiFISH for the segmentation genes *even-skipped* (*eve*) and *engrailed* (*en*) are shown in early and later embryos from five different arthropod species, *Drosophila melanogaster* (*D.mel*), *Drosophila virilis* (*D.vir*), *Tribolium castaneum* (*T.cas*), *Nasonia vitripennis* (*N.vit*) and *Parhyale hawaiensis* (*P.haw*). Embryos are oriented with anterior to left. smiFISH for *wingless* (*wg*) and *en* is shown in the *D.mel* imaginal wing disc. Ovaries were stained for the maternally loaded RNAs *bicoid* (*bcd*) and *nanos* (*nos*), which accumulate at the anterior and posterior poles of the developing egg, respectively. A single stage 10 egg chamber is shown, oriented with nurse cells and the anterior of the developing egg to left. DAPI was used to stain cell nuclei. All images were acquired using a white light laser scanning confocal microscope with 40X or 100X objectives. White dashed boxes are magnified to the right. Single mRNAs are visible for all samples tested.

multiple distinct gene expression patterns at single molecule resolution. We were able to separate nine colours; eight *Drosophila* Hox genes at single molecule resolution together in the same embryo, plus DAPI to stain nuclei (Fig. 2). Probe/fluorophore combinations are supplied in Supplementary Data 2. The image is provided as a high-resolution Supplementary Image 1 (available at https://github.com/LliliansCalvo/smiFISH_Arthropods), where both transcriptional sites, and single mRNAs can be observed with

zoom. Long genes tend to show large bright transcriptional sites, representing a localized accumulation of multiple nascent RNAs in the process of transcription along the gene length, for example *Antp* (~103 kb) and *Ubx* (~78 kb). Depending on the question, it may be informative to determine nascent RNA number. This can be achieved by calculating the ratio of the transcription site intensity to single mRNA spot intensity, to infer polymerase occupancy[30].

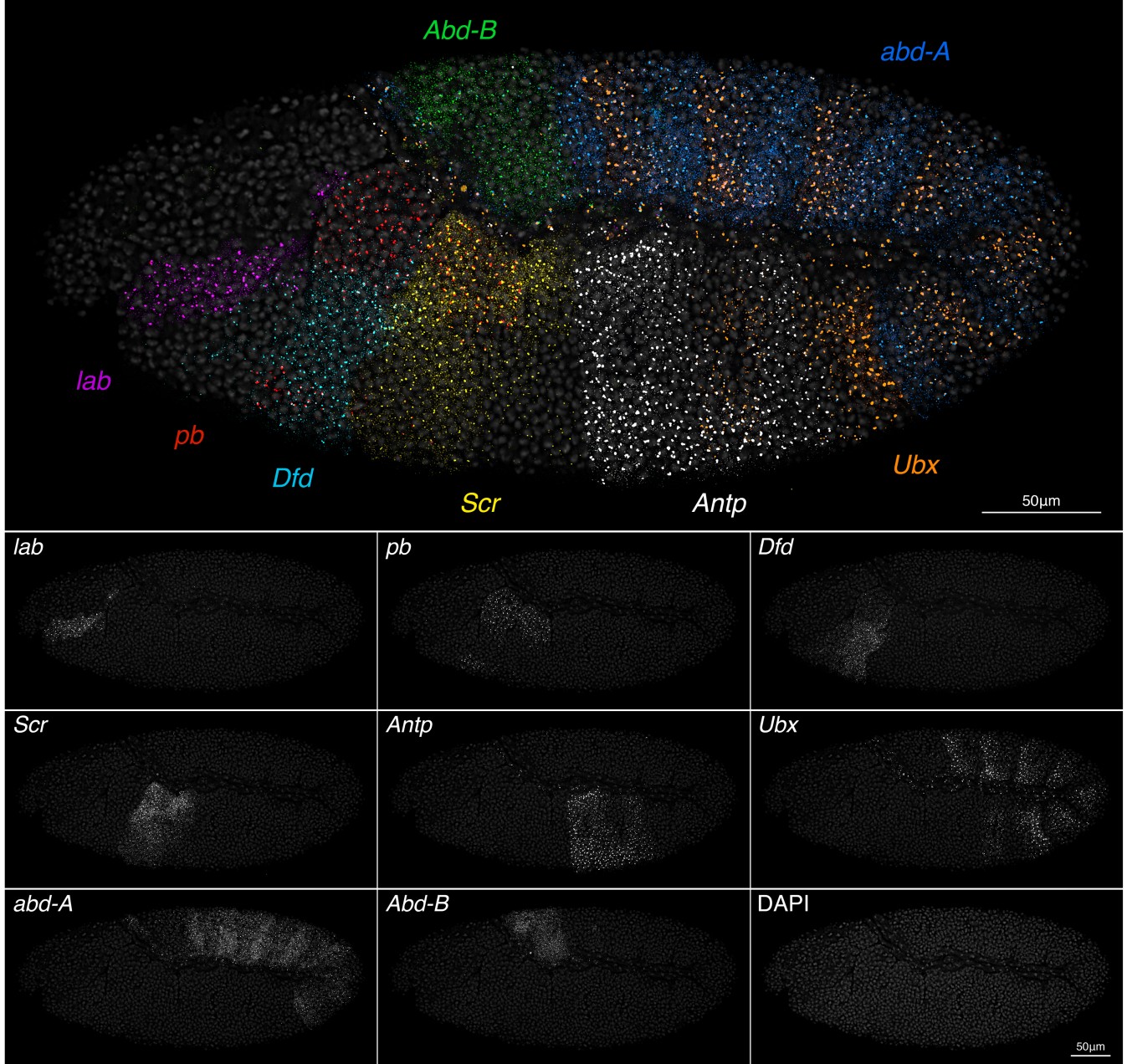

**Fig. 2 smiFISH and white light laser confocal imaging to visualize all eight *Drosophila* Hox genes at single molecule resolution.** A stage 10 germband extended *D. melanogaster* embryo (lateral view, anterior left) with smiFISH staining for all 8 Hox genes, plus DAPI to show the nuclei. The X-flap sequence was used for all probes, with the following fluorophores: *labial* CalFluor 610, *proboscipedia* Quasar 570, *Deformed* AlexaFluor 488, *Sex combs reduced* Quasar 670, *Antennapedia* promoter 1 CalFluor 540, *Ultrabithorax* Quasar 705, *abdominal-A* CalFluor 590 and *Abdominal-B* CalFluor 635. The image stack was acquired using a Leica SP8 inverted confocal microscope, with 40X objective and a white light laser, enabling optimal excitation wavelengths for each fluorophore. Peak emissions were captured by narrow ~20 nm tunable collection windows, and the image spectrally unmixed in Leica LAS X software to correct any residual bleed-through. Large bright spots mark accumulations of nascent RNAs at transcription sites; smaller, fainter spots are single mRNAs.

To view eight genes together, optimal excitation and collection from each fluorophore is essential to avoid bleed-through between channels. This image was acquired using a Leica SP8 confocal with white light laser, tunable to each specific excitation wavelength. Narrow collection windows of ~20 nm were set, corresponding to emission peaks of each fluorophore. Line averaging 16X, and high-resolution 4096 × 4096 format enabled single RNAs to be resolved. Despite settings that minimized bleed-through, some still persisted between certain channels, so the image was spectrally unmixed following acquisition. To avoid the need for spectral unmixing, a six colour stain using DAPI, AlexaFluor 488, Quasar 570, CalFluor 610, Quasar 670 and Quasar 705 is ideal.

**Whole-embryo segmentation for single-cell multi-gene RNA quantification.** The primary advantage of smFISH is to quantify RNA on a cell-by-cell basis, while preserving positional context. Distinguishing individual cells in culture is straightforward if spacing is sufficiently sparse, but in embryos or tissues is more challenging, and requires a cell membrane marker and

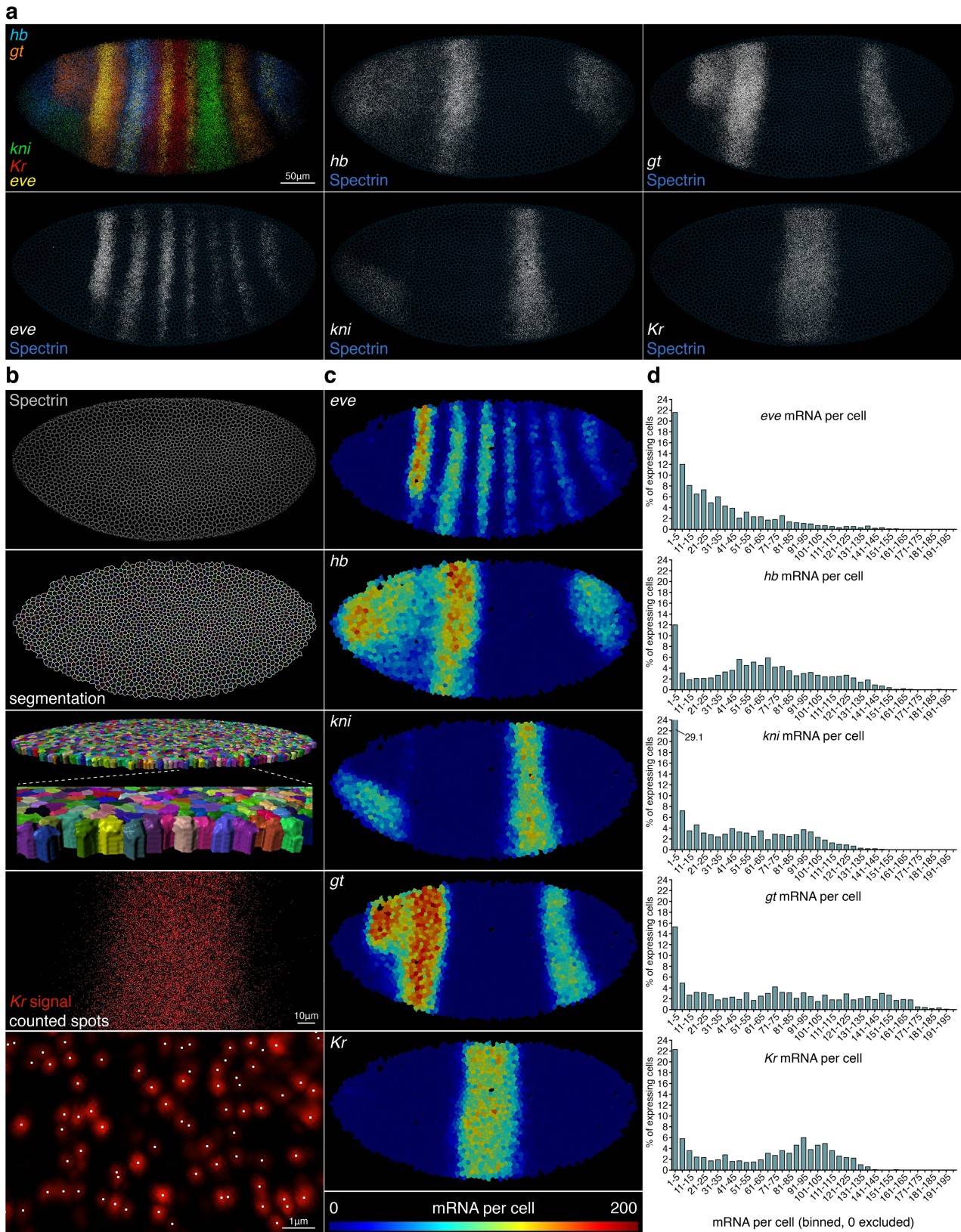

segmentation. We tested the compatibility of smiFISH with cell membrane immunofluorescence using a panel of different *Drosophila* antibodies, and found that immunofluorescence is best incorporated after smiFISH, not before. We identified alpha-Spectrin as an ideal marker that clearly defines cell boundaries and is least compromised by the prior smiFISH steps.

To quantify RNAs from multiple genes in single cells, we performed smiFISH in *Drosophila* embryos for four gap genes expressed at blastoderm stage—*hunchback* (*hb*), *giant* (*gt*), *knirps* (*kni*) and *Kruppel* (*Kr*), the pair rule gene *eve*, and marked cell membranes by Spectrin immunofluorescence (Fig. 3a). For probe/fluorophore combinations see Supplementary Data 2. Spectrin

**Fig. 3 smiFISH with membrane immunofluorescence allows whole-embryo 3D segmentation and multi-gene single-cell RNA quantification. a** Stage 5 cellular blastoderm *D. melanogaster* embryo (lateral view, anterior left) with maximum projections of smiFISH for the pair rule gene *even-skipped*, and four gap genes: *hunchback, knirps, giant* and *Kruppel*. Cell membranes are stained by immunofluorescence, using mouse anti-*Drosophila* alpha Spectrin, and goat anti-mouse Alexa Fluor 488. The confocal image stack comprises 48 slices at 200 nm z-intervals (9.6 μm total depth), from the apical limit of mRNA spots, to the basal extent of membrane ingression, to capture all RNAs that could accurately be assigned to single cells. **b** The cells module in Imaris v9.2 software was used to automatically segment Spectrin staining in 3D through the confocal stack, creating individual cell volumes (1982 in total). The Imaris spots module was used to automatically identify mRNA spots for each gene; and automatically assign spots to cell volumes based on *x, y, z* coordinates. **c** Heatmaps displaying mRNA number per cell for each of the five genes. **d** Histograms of mRNA number per cell for each gene, using bins of five with zero excluded. The shape of histogram distributions is a product of the expression patterns of the genes.

staining forms a clear cell border in z-slices where the cells are in cross section (Fig. 3b). In cellular blastoderm *Drosophila* embryos, cell membranes are in the process of ingressing between nuclei, but have not yet sealed off the basal side, causing Spectrin staining to fade out basally. mRNAs can be observed at z-planes beyond this basal membrane limit, but it is not possible to know with certainty which cell these basal mRNAs originated in. Therefore, for the purposes of cell-to-cell variability comparison, quantitation was limited to z-planes between the apical side and the basal limit of membrane ingression. For segmentation, a core set of an initial 20 z-slices (total of 4 μm) that do show clear cross-sectional Spectrin staining was identified, and the bottom slice of this core was then replicated to extend through an additional 28 slices (total of 5.6 μm) to the basal limit of membrane ingression where Spectrin staining disappeared. The cells module in Imaris software was used to segment the embryo in 3D through the full 48 slice (total depth 9.6 μm) z-stack, and the spots function to identify individual RNAs for each gene, which are then automatically assigned to cells (Fig. 3b). Details of Imaris analysis steps are provided in online Methods.

Heatmaps display the number of mRNAs of each gene, in each cell of the embryo up to the basal cell membrane ingression limit (Fig. 3c). These illustrate that all five genes show expression domains with graded, rather than sharp borders, consistent with the gap expression patterns being established in response to maternal morphogen gradients such as bicoid, within a syncytial embryo. Separate expression domains of the same gap gene show different overall expression levels, suggesting that transcriptional regulation varies with cell position. Histograms of single-cell data are shown in Fig. 3d (cells with zero RNA excluded). For cultured cells, the shape of histogram distributions of this type has been used to make inferences about promoter behaviour[31], based on an assumption that the promoter in each cell has a common behaviour shared throughout the cell population, leading to a certain signature evident from the distribution. For example, a promoter with high bursts of transcription followed by long off periods is expected to produce a distribution with a long tail to high values[31], similar to that found here for *eve*. However, it is important to note that such inferences cannot be made for non-ubiquitously expressed genes in whole embryos and tissues, as the assumption does not hold true. The distributions in Fig. 3d represent a mixed population of cells, where each gene shows a variety of different transcriptional behaviours, depending on spatial position within the embryo. The resulting variety in histogram shapes is therefore just a reflection of the different gene expression patterns, not the product of a common promoter behaviour. To illustrate this point, compare the histograms for *Kr* and *eve*. *Kr* is primarily expressed in a single broad stripe. The low proportion of cells with 15–65 RNA represents the swift spatial transition between very low-expressing edge cells and high-expressing cells within the stripe, while the bump between 65 and 130 corresponds to the large number of high-expressing cells within the stripe. In contrast, *eve* is expressed in seven

narrow stripes. Multiple stripes means more edges, so a high proportion of cells have intermediate RNA numbers, filling out the 6–65 bins, and a lower proportion of high-expressing cells in the centre of stripes, so loss of the bump between 65 and 130. The histogram shapes of these genes are therefore explained by their patterns, and are not an emergent property of a consistent promoter behaviour.

To verify the accuracy of spot detection, two interleaved sets each of 41 smiFISH probes against *Kr* mRNA were labelled in two different colours (CalFluor 610 and Quasar 570), imaged simultaneously and counted (Supplementary Fig. 2a, b). This confirmed high correlation in *Kr* mRNA/cell detected in each colour (Spearman's r = 0.99, P < 0.0001). To check the accuracy of spot quantification at 40X and the degree of undercounting due to mRNA crowding, *Kr* mRNAs were imaged in the same embryo first with 40X objective, then with 100X, and spots were quantified using the same detection threshold (Supplementary Fig. 2c, d). At both magnifications, visual inspection of Imaris spots confirmed detection of both strong and faint spots, and successful separation of closely touching spots (Supplementary Fig. 2c). A comparison of *Kr* mRNA/cell between the two magnifications showed a slight significant increase in detection rate at 100X, likely due to improved separation of very close proximity mRNAs (Supplementary Fig. 2d, 40X mean = 54, max = 136, 100X mean = 66, max = 142, non-zero cells only). For biological questions where absolute mRNA number is critical, or for genes with densely expressed mRNAs, imaging at 100X may therefore be advantageous. However, for lowly expressed genes, for nascent transcription site analysis, or where whole-embryo data is more critical than absolute numbers, then lower magnification 40X imaging is preferable.

The embryo shown in Fig. 3 was imaged and quantified only to the basal extent of membrane ingression, to ensure accurate assignment of mRNA spots to individual cells. To determine absolute mRNA numbers (but without assured accuracy of cell assignment) and to assess biological variability, we analysed *Kr* mRNA expression in an additional 12 blastoderm embryos through the full depth of smiFISH signal, beyond the extent of membrane ingression. mRNA spots were detected through a depth of up to 120 z-slices (total depth 24 μm) (Supplementary Fig. 2e, f). Embryos were sorted into age order by measuring the degree of membrane ingression between nuclei, determined from a cross-sectional plane in the Spectrin channel (Supplementary Fig. 2g), and *Kr* mRNA spots per cell were quantified using identical imaging and spot detection settings for all embryos (Supplementary Fig. 2h). *Kr* showed substantial biological variability between embryos, but not correlating with embryo age. Embryos showed maxima that ranged from 123 to 337 *Kr* mRNA/cell.

**Automated cell neighbour detection for cell-to-cell variability analysis.** To assess variability in gene expression, one can analyse the mean and spread of RNA values within the whole population

of cells, and compare individual cells to this distribution. However, since a given gene may show complex patterns comprising different domains expressing at different levels, analysing cells together as a single pool may not be informative. Single-cell variability is better addressed by comparing variability between a cell and its immediate neighbours. We define immediately neighbouring cells as those that directly share a membrane border in the Spectrin channel. Using a 2D segmentation plane from the embryo shown in Fig. 3b, immediate neighbour number of each cell was manually counted in half of the embryo (Fig. 4a), giving a range of 2–8, with the frequency distribution shown in Fig. 4d. To automate neighbour detection, the spots function in Imaris was employed with a low threshold to detect ~80,000 spots in the Spectrin channel, providing a dense representation of the membrane as spots (Fig. 4b). A custom R code was developed that uses both Spectrin spot coordinates and the cell ID to which each spot belongs, to construct 3D polygons that closely matched the original segmentation (Fig. 3c). Each polygon was slightly expanded in 3D by 0.6 μm (approximately 10% of a cell diameter), causing polygon boundaries to now intersect with just their immediate neighbours. The code then detects all intersections, producing for each polygon a list of directly neighbouring polygons (cells). The neighbour distribution obtained by this automated method closely matches the manual count distribution (Fig. 4d, e). To confirm agreement between manual and automated methods for specific cells, the neighbour number obtained with each method was compared for the first 200 cell IDs (Fig. 4f). Of the 200 cells, 87% had perfect agreement, and the maximum discrepancy between methods was ±1 neighbour.

The strength of this polygon method over a more simplistic neighbour search radius approach is that the polygon expansion factor is a small fraction (~10%) of the average cell diameter, therefore serves to exclusively identify only directly bordering cells, even in tissues with different cell sizes and shapes. For certain studies, comparing cells within a given area of tissue may be more meaningful than limiting to only touching cells. To provide this as an alternative option for the analysis pipeline, we also developed the code to calculate the Euclidian distance between the centre point coordinates of every cell, and then filter by a specified radius, to return for each cell a list of all neighbours within that radius. All the code is provided via the link under 'Code availability'.

**New measures to capture numerical and proportional single-cell variability.** The smiFISH panels in Fig. 5b show *eve*-expressing cells from an early germband *Parhyale* embryo, and highlight how a single cell can have a markedly different expression level from its immediately adjoining neighbours. Such single-cell variability within a population has been shown to have important biological relevance, for example in fate determination[11], cell behaviour[10] and disease[12]. Fano factor is a commonly used measure of local mRNA variability, and is calculated as variance/mean (Fig. 5a). Variance and mean are population measures, so all cells in the group are assigned the same Fano factor value, the resolution of which is therefore dictated by the size of the neighbour group. However, Fano factor cannot distinguish single variable cells *within* the neighbour group. This is illustrated by comparing the three hypothetical scenarios depicted in Fig. 5 c–e. The centre cells in panels (c) and (e) are equally different from their neighbours, both proportionately and numerically, whereas the centre cell in panel (d) is not very variable, being the same as all but one of its neighbours. However, Fano factor fails to distinguish any difference between (c) and (d) (both 6.11) and incorrectly finds (e) much more

variable than (c) (42.37 vs 6.11). To overcome this limitation, we devised two alternative variability measures, the local numerical cell variability (NV) and the local proportional cell variability (PV) (Fig. 5a). Both measures express how different an individual cell is from its immediate neighbours. NV is normalized by the maximum mRNA per cell for the whole cell population, therefore a high NV value highlights the cells whose mRNA value difference with their immediate neighbours is numerically large in terms of the maximum level at which that gene can be expressed. PV is normalized by the maximum just for the neighbour group, and so high PV does not necessarily mean a large difference in actual mRNA number, just that the cell has a high proportional difference from its neighbours. Both measures return values between 0 (no variability) and 1 (maximum variability). In the scenarios shown in Fig. 5 c-f, 550 is used as the population maximum. Both NV and PV find the centre cells in scenarios (c) and (e) to be equally variable, and (d) to be less variable. Importantly, NV is the same between scenarios (c), (e) and (f), since the numerical RNA difference between the centre cell and each neighbour is the same, whereas PV finds scenarios (c) and (e) to be proportionally more variable than (f).

Fano factor, NV and PV were calculated for the five genes shown in Fig. 3, using neighbours defined by the automated neighbour detection method (Fig. 4). Variability scores are displayed as heatmaps (Fig. 5g). Cells outside of expression domains that have a single mRNA, surrounded only by non-expressing neighbours, have the maximum PV score of 1. While this is correct, we were more interested to highlight cells that had high PV within actual expression domains. Therefore, when calculating PV, cells were filtered on the criteria of neighbour-group mean ≥1; cells failing this criterion were assigned a score of 0. The heatmaps show how Fano factor, NV and PV highlight different aspects of variability. Fano factor picks out the edges of expression domains. It acts like a moving average variability, and therefore highlights the regions (but not individual cells) where RNA number is changing the most with position. Within the centre of expression domains, Fano factor is generally low, suggesting a similar expression level. In contrast, NV can highlight individual cells within the centre of domains that have a high difference in mRNA number from neighbours; the cells that were overlooked by Fano factor. For example, contrast NV and Fano factor for *kni* and *Kr*. PV highlights cells that are proportionately most different from neighbours, which tends to be cells at the extreme edges of domains, at the transition between off and on. However, individual cells with high PV can still be observed throughout the expression domains of each gene.

## Discussion

Whole-genome DNA and RNA sequencing is becoming increasingly feasible and affordable, and consequently the number of non-model organisms with whole or partial genome sequence is rapidly growing. Since only ~1 kb of gene sequence is required to design a probe set, smFISH can be applied with ease to non-model species, revealing both expression patterns and levels. Here we have tested smiFISH, with modifications, across a range of arthropod species and sample types, and found that it enabled single mRNA visualization with consistency and high specificity. We also combined smiFISH with subsequent membrane immunofluorescence, allowing whole-embryo single-cell segmentation. The anti-*Drosophila* alpha-Spectrin antibody used did not work in the non-*Drosophilid* species tested, so appropriate species-specific membrane antibodies are required for use in different organisms.

smiFISH makes multiplexing simple and flexible, and therefore imaging becomes the limitation on how many genes

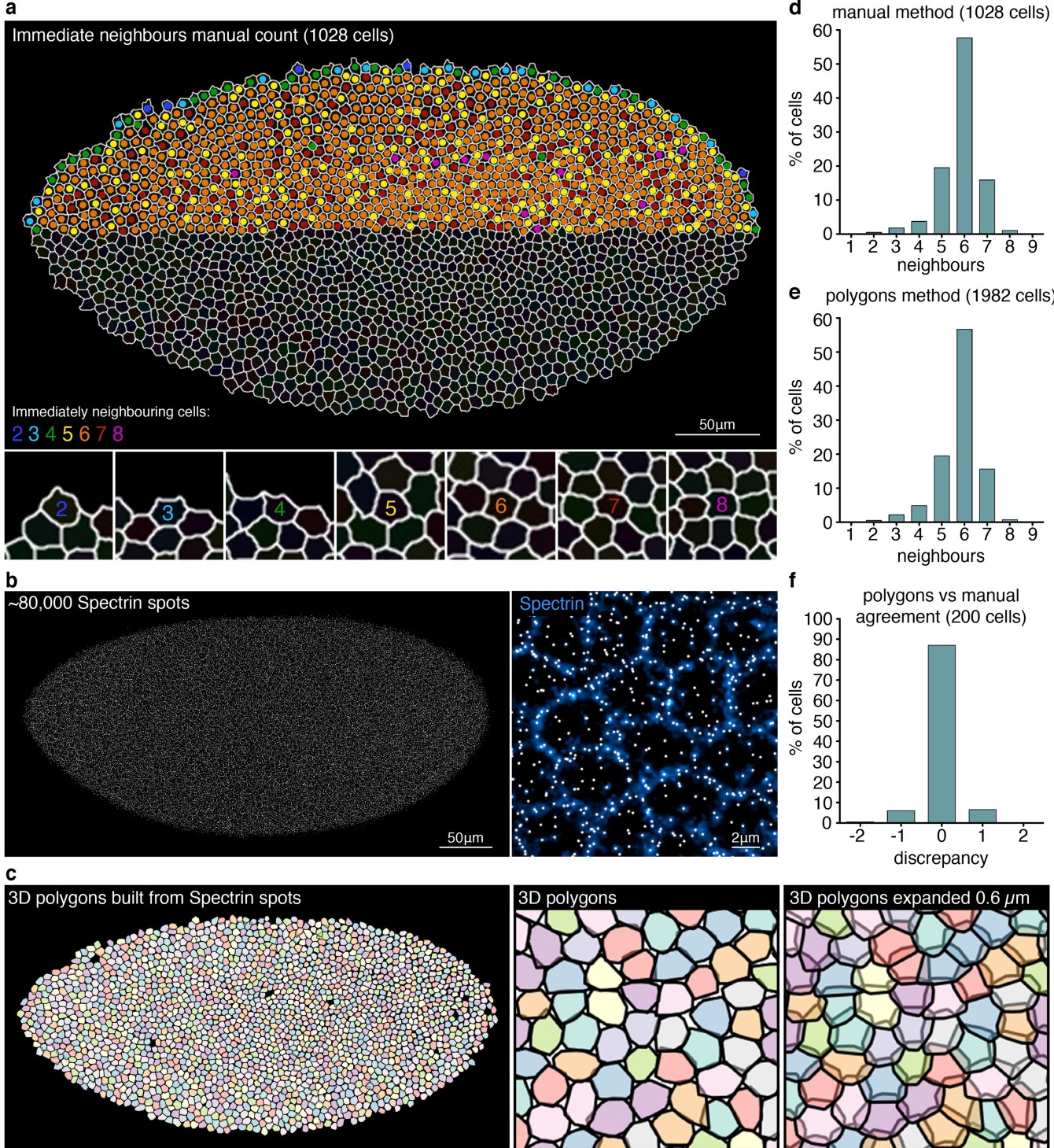

**Fig. 4 Automated identification of immediately neighbouring cells for single-cell variability analysis. a** 2D segmentation plane from a cellular blastoderm *D. melanogaster* embryo with anti-Spectrin membrane staining. The number of immediately neighbouring cells (defined as directly sharing a portion of membrane) was manually counted, for each cell in half of the embryo. A range of 2–8 immediate neighbours was found. **b** Low-threshold Imaris spot detection in the Spectrin channel, to generate dense representation of the membrane with spots, for 3D polygon generation. **c** A custom R code was developed to generate 3D polygons from Spectrin spot coordinates and assigned cell IDs, closely reproducing initial segmentation. Each polygon was slightly expanded in 3D by 0.6 μm (~10% of a cell diameter), generating intersections between borders of only directly neighbouring cells. The code detects all intersections, producing for each polygon a list of directly neighbouring cells. **d** Histogram summarizing the manual neighbour count (1028 cells) in (**a**). **e** Histogram summarizing the automated neighbour count (1982 cells) in (**c**), confirming close agreement with the manual count. **f** Histogram summarizing a direct cell-by-cell comparison of neighbour number between the manual count and the automated method, for the first 200 cell IDs, calculated as manual minus automated.

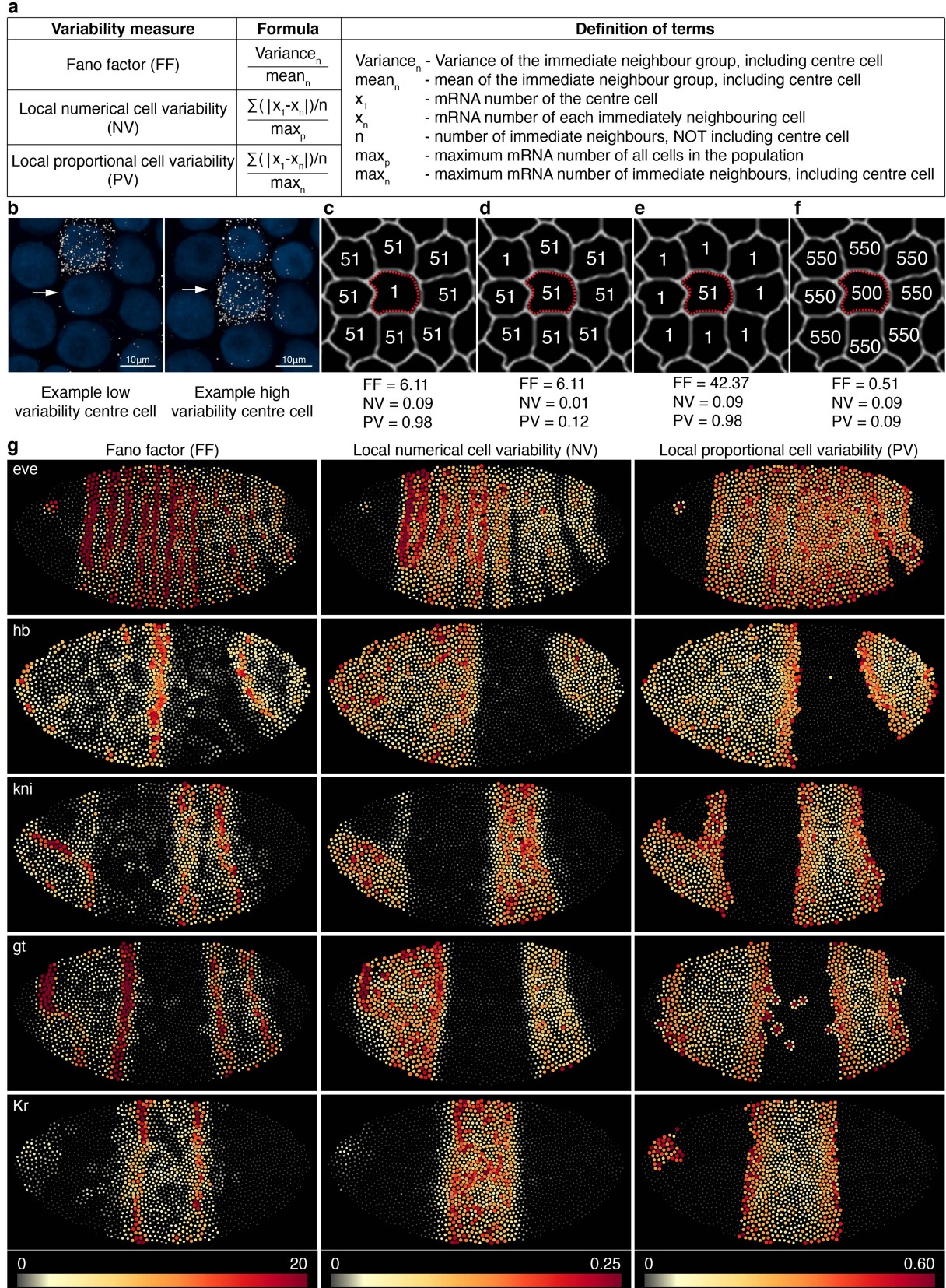

**a**

| Variability measure | Formula | Definition of terms |
|---|---|---|
| Fano factor (FF) | $\dfrac{Variance_n}{mean_n}$ | $Variance_n$ - Variance of the immediate neighbour group, including centre cell<br>$mean_n$ - mean of the immediate neighbour group, including centre cell |
| Local numerical cell variability (NV) | $\dfrac{\sum(|x_1-x_n|)/n}{max_p}$ | $x_1$ - mRNA number of the centre cell<br>$x_n$ - mRNA number of each immediately neighbouring cell<br>$n$ - number of immediate neighbours, NOT including centre cell |
| Local proportional cell variability (PV) | $\dfrac{\sum(|x_1-x_n|)/n}{max_n}$ | $max_p$ - maximum mRNA number of all cells in the population<br>$max_n$ - maximum mRNA number of immediate neighbours, including centre cell |

**b** Example low variability centre cell Example high variability centre cell

**c**
FF = 6.11
NV = 0.09
PV = 0.98

**d**
FF = 6.11
NV = 0.01
PV = 0.12

**e**
FF = 42.37
NV = 0.09
PV = 0.98

**f**
FF = 0.51
NV = 0.09
PV = 0.09

**g** Fano factor (FF) Local numerical cell variability (NV) Local proportional cell variability (PV)

eve, hb, kni, gt, Kr

can be viewed together. Using an imaging strategy to optimize fluorophore excitation and capture of emission peaks, we could image nine different channels simultaneously with spectral unmixing, or six channels without unmixing.

The capacity to image more genes simultaneously is advantageous as it allows more potentially interacting genes to be studied within the same cell, thus eliminating error due to sample variability.

**Fig. 5 Three alternative measures to express cell-to-cell mRNA variability. a** Formulae for Fano factor, a commonly used measure of cell variability, and NV and PV, two alternative measures designed to better capture individual cell variability. **b** Cells from the anterior stripe of *eve* expression in *P. hawaiensis* early germband embryo, highlighting that individual cells can differ greatly in expression from their immediate neighbours. The centre cell (white arrow) in the left panel is similar to all its neighbouring cells except one, whereas the centre cell in the right panel is highly different from all of its neighbours except one. **c–f** Hypothetical scenarios of neighbour-group mRNA variability, to highlight the capacity of each formula to capture the variability of the single centre cell (red dashed border). Here, 550 mRNA/cell is used as the population maximum. Fano factor incorrectly returns the same value for (**c**) and (**d**), and incorrectly finds (**e**) to be more variable than (**c**). NV correctly returns the same value for (**c, e** and **f**), and a low value for (**d**). PV correctly returns the same value for (**c**) and (**e**), and lower values for (**d**) and (**f**). **g** Heatmaps show the three different variability measures, calculated for each segmented cell of the embryo (1982 cells in total), for five genes: *even-skipped, hunchback, knirps, giant* and *Kruppel*. Dots representing cells are scaled in both size and colour by the variability value. Both NV and PV measures can range from minimum 0 to maximum 1. For PV heatmaps, cells were filtered on the criteria of neighbour-group mean mRNA number $\geq$1; cells failing this criterion were assigned a score of 0.

A major strength of smFISH is that position of the cell within the sample is preserved, which allows variability to be analysed on a cell-by-cell basis. We compared a commonly used measure of cell variability, the Fano factor, with two alternative measures termed NV and PV that were devised to better highlight individual cell variability. Each measure has its own strengths and limitations, and therefore is appropriate for different applications. The Fano factor highlights regions where the RNA number is changing most with cell position, but was not capable of comparing individual cells to their immediate neighbours. In contrast, NV was effective at highlighting individual cells that were markedly different numerically in mRNA from their neighbours. NV is therefore a relevant measure for questions where the absolute RNA number is important, for example when a threshold expression level is required for a particular process to occur[32,33], or post-transcriptional buffering mechanisms that fine-tune mRNA levels[34,35]. PV also effectively highlighted individual cells that differed from their neighbours, but in proportional expression rather than actual. The PV measure is most relevant for questions involving the mechanisms of RNA production, such as promoter dynamics, and the effects of enhancers and transcription factors. Large PV values may indicate fundamentally different transcription dynamics between cells. This is not necessarily true of high NV, which could be attained in a region of high-expressing cells all displaying the same fundamental promoter behaviour, but with some stochasticity that causes a proportionally small, but numerically large RNA difference between cells.

Depending on the biological question, statistics can subsequently be performed on the NV and PV values. For example, cells with a NV or PV score significantly higher than the population mean score can be identified. We envisage that this approach may be useful in cancer studies to highlight, within a tissue, the most abnormally variable cells with respect to their neighbours. Similarly, PV and NV can be compared statistically between different genotypes or experimental conditions to assess how specific factors affect expression variability. For example, the effect of microRNAs on buffering mRNA fluctuations could be assessed by comparing PV and NV between samples with and without the presence of microRNA.

In summary, this work provides a straightforward methodology applicable across a variety of different animal systems, enabling in-depth molecular analyses that traditionally were only feasible in established model systems. Our analysis pipeline to obtain single-cell RNA counts in whole embryos is relevant for studying diverse aspects of expression analysis, and we anticipate that the detailed multi-colour imaging strategy provided here will prove valuable for analysis of gene networks. Finally, it is our view that methods to appropriately analyse spatial cell-to-cell variability will yield a new level of information critical to understand how individual cell behaviours lead to biological outcomes.

## Methods

**Solutions**. For dechorionation: 50% bleach: 50% sodium hypochlorite solution in distilled $H_2O$. Embryo wash buffer: 0.1 M NaCl, 0.02% Triton X-100 in distilled $H_2O$. Fix solution: 0.5 ml 10X PBS (Sigma, without $CaCl_2$ and $MgCl_2$), 0.5 ml nuclease-free $H_2O$, 4 ml ultrapure 10% methanol-free formaldehyde (Polysciences), 5 ml heptane 99% (Sigma). PBT: 1X PBS (Sigma, without $CaCl_2$ and $MgCl_2$), 0.05% Tween 20 (Sigma), in nuclease-free $H_2O$. smiFISH wash buffer: 2X SSC, 10% deionized formamide (Ambion) in nuclease-free $H_2O$. smiFISH hybridization buffer: 10% w/v dextran sulphate (Sigma, molecular weight 6500–10,000), 2X SSC, 10% deionized formamide in nuclease-free $H_2O$. Blocking solution: 1X Western Blocking Reagent (Sigma) in PBT.

**Sample fixation**. *Drosophila* embryos collected on apple juice agar plates at 25 °C were dechorionated with 50% bleach, alternately washed with distilled water and embryo wash buffer, and shaken in fix solution at 240 rpm for 45 min. The aqueous solution layer was removed, 10 ml 100% methanol was added, and shaken for 1 min to devitellinize embryos. Devitellinized embryos were washed 5X with 100% methanol, then stored in 100% methanol at −20 °C. Fixed *Tribolium* embryos were kindly provided by Olivia Tidswell from Michael Akam's lab. *Tribolium* were dechorionated and fixed as described for *Drosophila*, but for increased devitellinization efficiency, embryos were passed through a 19G needle in ice-cold 100% methanol. Fixed *Nasonia* embryos were kindly provided by Shannon Taylor from Peter Dearden's lab[36]. *Parhyale* embryos were manually collected from females anaesthetized with 0.01% clove oil in sea water. Embryos were washed 3X with filtered sea water, transferred to fix solution, and shaken at 240 rpm for 45 min. Embryos were then transferred to a glass dish in 1X PBS, for manual removal of the chorion and vitelline membrane using tungsten needles. Dissected embryos were transferred to a second fixation solution of 4% formaldehyde in 1X PBS for 1 h, before washing 5X in 100% methanol and storage in 100% methanol at −20 °C. For imaginal discs, white pre-pupae were chilled in 1 °C 1X PBS, the cuticle opened longitudinally, and pupae fixed in a solution of 4% formaldehyde in 1X PBS for 1 h (no rocking), before washing 5X in 100% methanol and storage in 100% methanol at −20 °C. For ovaries, adult females were thoroughly anaesthetized with $CO_2$, placed in 1 °C 1X PBT, and ovaries dissected out and opened to expose ovarioles. Dissected ovaries were washed with 1X PBS, then fixed and stored as described above for pupae.

**Probe design and preparation**. *D. melanogaster* and *D. virilis* mRNA sequences were obtained from Flybase (https://flybase.org); *T. castaneum, N. vitripennis* and *P. hawaiensis* mRNA sequences were obtained from NCBI (https://www.ncbi.nlm.nih.gov/nucleotide/). Complementary 20 nt DNA probes against mRNA sequences (up to 48 probes per gene) were designed using the Biosearch Technologies stellaris RNA FISH probe designer tool (free with registration, https://biosearchtech.com). All probe sequences are provided in Supplementary Data 1. The following sequence was added to the 5′ end of each 20 nt probe: CCTCCTAAGTTTCGAGCTGGAC TCAGTG. This is the reverse complement of the X FLAP sequence used in Tsanov et al.[16]. Oligos were ordered from Integrated DNA Technologies (IDT), in 96-well plate format, using 25 nmol synthesis scale, standard desalting, and at 100 μM in nuclease-free $H_2O$. A standard volume (we use 50 μl) of each probe was combined together to generate an equimolar probe mix, at 100 μM (mixed probe concentration). A set of 48 probes therefore gave 48 × 50 μl = 2.4 ml of probe mix, which was split into 100 μl aliquots for storage at −20 °C. The X FLAP sequence itself CACTGAGTCCAGCTCGAAACTTAGGAGG was 5′ and 3′ end-labelled with CalFluor 540, Quasar 570, CalFluor 590, CalFluor 610, CalFluor 635, Quasar 670 and Quasar 705, and synthesized by Biosearch Technologies. X FLAP sequence 5′ and 3′ end-labelled with Alexa Fluor 488 was synthesized by IDT. Lyophilized fluorophore-labelled FLAP was resuspended in nuclease-free $H_2O$ to a concentration of 100 μM, aliquoted and stored at −20 °C.

**Probe FLAP annealing**. Probes were annealed to fluorophore-labelled FLAP sequences in 50 μl reactions as follows:

| Component | Volume per reaction | Amount |
|---|---|---|
| Probe set (100 μM mixed probe concentration) | 2.0 μl | 200 pmol (total DNA) |
| Fluorophore-labelled FLAP (100 μM) | 2.5 μl | 250 pmol |
| 10X NEB 3* | 5.0 μl | |
| Nuclease-free H$_2$O | 40.5 μl | |

*NEB 3: New England Biolabs Buffer 3 (1X composition: 100 mM NaCl, 50 mM Tris-HCl, 10 mM MgCl$_2$, 1 mM DTT, pH 7.9).

When more than three probe sets were to be used on the same sample, probes were annealed at 5X concentration, so they could be used at 1/5 normal volume in the hybridization buffer, to avoid large volumes of probe affecting salt and formamide concentrations in the hybridization. The 5X concentration annealing reactions were set up as follows:

| Component | Volume per reaction | Amount |
|---|---|---|
| Probe set (100 μM mixed probe concentration) | 10.0 μl | 1 nmol (total DNA) |
| Fluorophore-labelled FLAP (100 μM) | 12.5 μl | 1.25 nmol |
| 10X NEB 3* | 5.0 μl | |
| Nuclease-free H$_2$O | 22.5 μl | |

Annealing reactions were performed in a thermal cycler according to Tsanov et al.: Lid 99 °C, 85 °C 3 min, 65 °C 3 min, 25 °C 5 min, 4 °C hold[16]. Final annealed probes are at either 4 μM (from the normal concentration annealing reaction) or 20 μM (from the 5X concentrated annealing reaction). Annealed probes were stored at −20 °C. Probe/fluorophore combinations are supplied in Supplementary Data 2.

**smiFISH and immunofluorescence**. All steps were performed in glass Wheaton V-Vials (Sigma). Using glass is important because embryos and tissues stick to plastic in the smiFISH wash buffer and hybridization buffer, and the v-base helps with sample visualization and retention through the multiple solution changes. Fixed samples stored in 100% methanol were transitioned to PBT in stages: 50% PBT, 75% PBT, 100% PBT, 5 min per wash. Samples were washed 3 × 10 min in PBT, then 10 min in 50% PBT 50% smiFISH wash buffer, followed by 2 × 30 min pre-hybridization washes in smiFISH wash buffer at 37 °C. Annealed smiFISH probes (4 μM or 20 μM) were diluted in 500 μl smiFISH hybridization buffer to a concentration of 80 nM. Probes were hybridized with samples in the dark at 37 °C for 14 h. Samples were washed 4 × 15 min in smiFISH wash buffer at 37 °C, then 3 × 10 min in PBT at room temperature. For immunofluorescence, samples were blocked for 30 min in blocking solution, then incubated with anti-*Drosophila* alpha-Spectrin (DSHB 3A9) diluted 1:50 in blocking solution, for 18 h at 4 °C. Samples were washed 4 × 15 min with PBT, blocked for 30 min, incubated with goat anti-mouse Alexa Fluor 488 (ThermoFisher) diluted 1:500 in blocking solution, for 4 h at room temperature, then washed 4 × 15 min with PBT. In PBT, *Tribolium* and *Parhyale* embryos were manually dissected away from yolk using tungsten needles, imaginal discs were dissected away from pupal carcasses, and single ovarioles and egg chambers were dissected away from one another. All samples were mounted under #1.5 coverslips using ProLong Diamond Antifade Mountant with DAPI (ThermoFisher). Due to their size, coverslip spacers were required for *Parhyale* embryos.

**Imaging**. Images were acquired on a Leica TCS SP8 AOBS inverted gSTED microscope using a 40X/1.3 or 100X/1.4 HC PL APO (oil) objective. Image stacks for each different species, imaginal discs and ovaries were taken with the following settings: format 2048 × 2048 or 4096 × 4096, speed 400 Hz unidirectional, sequential line scanning, line averaging 8 or 16, pinhole 1 airy unit. Each channel was gated 1.0–6.0. DAPI excitation 405 nm, laser 5%, collection 415–480 nm. CalFluor 610 excitation 590 nm, laser 20%, collection 600–642 nm. Quasar 670 excitation 647 nm, laser 20%, collection 657–750 nm.

*D. melanogaster* embryo image stacks showing all 8 Hox genes (Fig. 2), 5 segmentation genes (Fig. 3) or *Kr* mRNA for quantification validations (Supplementary Fig. 2) were taken with the following settings: format 4096 × 4096, speed 400 Hz unidirectional, sequential line scanning, line averaging 16, pinhole 1 airy unit. Each channel was gated 1.0–6.0. DAPI excitation 405 nm, laser 5%, collection 415–480 nm. AlexaFluor 488 excitation 490 nm, laser 15%, collection 498–530 nm. CalFluor 540 excitation 522 nm, laser 15%, collection 530–555 nm. Quasar 570 excitation 548 nm, laser power 15%, collection 558–575 nm. CalFluor 590 excitation 569 nm, laser 15%, collection 579–595 nm. CalFluor 610 excitation 590 nm, laser 15%, collection 605–620 nm. CalFluor 635 excitation 618 nm, laser

15%, collection 628–650 nm. Quasar 670 excitation 647 nm, laser 10%, collection 660–680 nm. Quasar 705 excitation 670 nm, laser 10%, collection 695–780 nm. Image stacks were acquired with a 200 nm *z*-interval. For images intended for mRNA quantification, *z*-stack limits were set to start just above the most apical smiFISH signal, and to end either at the basal extent of membrane ingression (to capture all RNAs that can accurately be assigned to single cells), or just below the basal extent of smiFISH signal (to capture all mRNAs throughout the full cytoplasmic depth). Spectral unmixing of the 8 Hox gene channels was performed in the Leica LAS X v1.8.0.13370 software, using a 30 μm radius selection in each channel to build the unmixing matrix.

**Image analysis**. *z*-stacks were stabilized through *z* to account for any imaging drift, and deconvolved using Huygens Professional v18.04. For single-cell segmentation and mRNA quantification, DAPI, Spectrin and smiFISH image stacks were combined in Imaris v9.2 software. Spectrin staining forms a clear cell border in *z*-slices where the cells are in cross section, but fades out basally at the extent of membrane ingression. The core set of *z*-slices that do show clear cross-sectional Spectrin staining was identified, and the top and bottom slices of this core replicated to extend through the full depth of the stack, replacing apical and basal slices with unclear cell borders. Cells were then segmented in 3D automatically in the Imaris cells module from the Spectrin channel, using a smallest cell diameter of 5 μm, membrane detail level of 0.5 μm and a local contrast filter. Edge cells and any double cells were omitted by filtering the set of detected cells for outliers based on cell volume, sphericity and *z*-position. smiFISH spots were detected using the spots function, allowing for different spot sizes, with an estimated *xy* diameter of 0.3 μm, estimated *z* diameter of 0.6 μm and background subtraction. Spot quality thresholds were set individually for each channel, since brightness and diameter of spots is inherently different between different fluorophores. Thresholds were set just below the point at which a sharp spike in background false positives outside of the established domain occurs. For quantitation consistency, the same detection thresholds were used for embryo comparisons. To prepare images for figures, maximum projections of smiFISH channels were generated in FIJI v2.0.0-rc-49/ 1.51d. Projections were then combined into RGB images in Adobe Photoshop CS6.

**Statistics and reproducibility**. Statistics are not used within the main figures, but all cell *n* numbers are defined in the figure legends.

Details of the statistical analysis performed in Supplementary Fig. 2, and all cell *n* numbers, are provided in the legend. Heatmaps displaying mRNA/cell were generated in Imaris v9.2 software. Cell variability heatmaps were generated in R using the package ggplot2. Statistical analyses and graphs were completed using GraphPad Prism v5.0.

**Reporting summary**. Further information on research design is available in the Nature Research Reporting Summary linked to this article.

## Data availability
All smiFISH probe sequences are available in Supplementary Data 1. All smiFISH probe/ fluorophore combinations are available in Supplementary Data 2. A high-resolution image showing smiFISH staining for all eight *Drosophila* Hox genes is provided as Supplementary Image 1. These supplementary files, and all data underlying the graphs and heatmaps presented can be accessed at https://github.com/LliliansCalvo/ smiFISH_Arthropods.

## Code availability
All custom code (R scripts) for automated detection of neighbouring cells and heatmap generation can be accessed at: https://github.com/LliliansCalvo/smiFISH_Arthropods and is deposited in zenodo[37].

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

## Acknowledgements

Special thanks to Olivia Tidswell from Michael Akam's lab, and to Shannon Taylor from Peter Dearden's lab, for kindly providing *Tribolium* and *Nasonia* embryos, respectively, and to Matthew Benton for advice with *Tribolium*. We also thank Peter March from the Manchester bioimaging facility for all his support, and Hilary Ashe, Sam Griffiths-Jones and Pauline Rogers for valuable comments and discussions. Attendance to the summer embryology course at Marine Biological Lab, Woods Hole, encouraged the conception of this work, so we are grateful to Rich Schneider and Dave Sherwood for their work in running the course. This work was supported by the Biotechnology and Biological Sciences Research Council (grant number BB/P002153/1) to M.R. and T.P., and a Wellcome Trust Ph.D. studentship (203990/Z/16/A) to L.C.

## Author contributions

Conceptualization L.C., M.R. and T.P.; experiments L.C. and T.P.; imaging and image analysis T.P.; analysis formulae L.C. and T.P.; coding and data analysis L.C.; writing – original draft L.C. and T.P., writing – review and editing L.C., M.R. and T.P.; funding acquisition M.R. and T.P.

## Competing interests

The authors declare no competing interests.
