## [Peer Review File · Communications Biology]

Reviewers' comments:

Reviewer #1 (Remarks to the Author):

In this manuscript, Calvo et al. describes applications of an established smFISH approach (smiFISH or single molecule inexpensive FISH) in several extant and emerging arthropod model organisms. They also developed image analysis pipelines for single-cell transcript expression analysis in embryos or tissues. In general, the data is of high quality, the text is clear, the method is detailed enough, and the analysis methods can be broadly interesting. Although there are no novel biological findings, that does not seem to be the focus of this manuscript. I recommend its publication if the following points can be addressed.

Major point:

The current neighbor-finding algorithm relying only on cell-cell distance is almost too simple and can only work for tissues with mostly near hexagon cells. In the case of cell shape changes during epithelial folding, for instance, some cells will be elongated, and the algorithm will clearly fail to faithfully identify neighbors.

For this type of analysis to be more broadly applicable, I'd recommend implement a more sophisticated algorithm. For example, one could start by a clearly large search radius, and determine how many edges the connecting line between the queried cell center and a potential neighbor cell center crosses; a true neighbor should cross only once, as long as the cell topology is not too complicated. A more robust algorithm will be identifying the vertices and edges of each cell to literally using shared edges as the criterion.

It is up for the editor and the authors to decide whether this is too much additional work, but I believe it is reasonable giving the coding capability gauged from the figures.

Minor points:

1. When describing eve expression in Figure 1, do authors suggest "low" eve mRNA expressing cells have post-transcriptional regulation that further sharpen protein-level expression boundaries, or just to highlight the superb smFISH resolution comparing to immunofluorescence and traditional in situ?
2. When segmenting cells using Spectrin signal, how far was the sharp mid-section Spectrin signal was extrapolated? Any supporting evidence or argument to support the choice of the cell height?
3. In Figure 1, posterior mRNA accumulation of the nos mRNA is marginal. Is this common?
4. The abbreviation MYA (million years ago) needs to be explained when first mentioned.
5. In Figure 2, the Hox gene multiplexing. Are bright foci representing active transcription bursts? Is it still possible to count mRNA dots in these foci? When plotting the distribution of mRNA expressions of these genes with direct evidence of active transcription, does the histogram have a long tail? Discussion of this point can be combined with Figure 3 discussion.

Reviewer #2 (Remarks to the Author):

Calvo et al. have adapted an existing single molecule FISH method, previously developed for cultured cells, to whole mount preparations. They demonstrate that the method works for embryos and tissues from a handful of arthropods. Using deconvolution and spectral unmixing, they show that up to eight genes can be assessed in whole mounts. They present a membrane labeling protocol compatible with their FISH method. They also present two straightforward measurements of cell-to-cell variability.

The mRNA labeling method will certainly be regarded by the community as a valuable cost-saving measure. The compatibility with membrane labeling will also be very useful for cell segmentation. Demonstrating the application of spectral unmixing is also useful. These are valuable additions to the toolbox of single mRNA molecule detection. However, the quantification is not rigorous and detracts from the manuscript. The manuscript otherwise contains no novel conclusions. The work might be more appropriate for a methods-centric publication once the authors address concerns about the quantification and measurement error outlined below.

To advance the use of single molecule FISH in whole mount tissues, it is essential to present the evidence that the procedure and subsequent analysis correctly measure mRNA density. But the mRNA counts per cell presented here are not consistent with previous studies. Further, to ascertain the extent of biological variability requires accounting for measurement error, but this has not been done. Sources of measurement error that the authors should account for include:

-Detection failure. The authors should estimate the extent of under-counting using two sets of probes labeled with different colors and directed against the same mRNA. Moreover, at a glance the failure rate appears quite high in the case of nos mRNA, single mRNAs of which are densely distributed through the entire oocyte (PMID: 9895314, 25848747).

-mRNA crowding. mRNAs will be undercounted at a rate that increases with increasing density. This is not accounted for in the manuscript, but has a major impact on any conclusions about variability. The authors do appear to be undercounting the gap gene mRNAs. For example, Kr has a maximum density of around 1000 mRNA per cell, where the basal extent of the cell is taken at 12 microns below the cortex (PMID: 23953111).

-Tissue deformation upon processing and mounting. The fixing and mounting procedure can alter the apparent density of mRNAs and nuclei and can be a dominant source of measurement error (PMID: 23953111, Fig. S2). In the current manuscript, such deformation might well contribute to measurement error, given the clear spatial correlations in the variability shown in Fig. 5G. For example, Kr variability approaches the Poisson limit in the most rapidly expressing regions of the embryo, i.e. Fano factor = 1 (PMID: 23953111, Fig. S6); however, in the current manuscript, the Fano factor clearly exceeds 1 in this region, and the Fano factor exhibits spatial correlations around 4-5 nuclear diameters. This is consistent with local tissue deformations impacting the counts per cell.

As a final point, the usefulness of the two new measurements of variability is not immediately clear. As a means of assessing variability between cells, it might be more informative to assign each cell a value based on how many standard deviations away from the "neighborhood mean count" that cell is found, and a p value to say whether that distance is significantly larger than expected. None of the

measurements presented (including the Fano factor) help the reader understand whether the mRNA content of a given cell is significantly larger (or smaller) than expected.

General note to Reviewers: To accompany figure changes and a new figure, there have been substantial changes made to portions of the text, and also some minor text changes throughout the document to fit with the points addressed. Therefore for clarity, rather than include all the text changes along with our responses, we have appended to the end of this document the full revised manuscript, including all figures. All changes are highlighted. For ease of locating the key changes, we indicate at the end of each reply the relevant figure or line number in the manuscript.

Reviewers' comments:

Reviewer #1:

In this manuscript, Calvo et al. describes applications of an established smFISH approach (smiFISH or single molecule inexpensive FISH) in several extant and emerging arthropod model organisms. They also developed image analysis pipelines for single-cell transcript expression analysis in embryos or tissues. In general, the data is of high quality, the text is clear, the method is detailed enough, and the analysis methods can be broadly interesting. Although there are no novel biological findings, that does not seem to be the focus of this manuscript. I recommend its publication if the following points can be addressed.

Major point:

The current neighbor-finding algorithm relying only on cell-cell distance is almost too simple and can only work for tissues with mostly near hexagon cells. In the case of cell shape changes during epithelial folding, for instance, some cells will be elongated, and the algorithm will clearly fail to faithfully identify neighbors.

For this type of analysis to be more broadly applicable, I'd recommend implement a more sophisticated algorithm. For example, one could start by a clearly large search radius, and determine how many edges the connecting line between the queried cell center and a potential neighbor cell center crosses; a true neighbor should cross only once, as long as the cell topology is not too complicated. A more robust algorithm will be identifying the vertices and edges of each cell to literally using shared edges as the criterion.

It is up for the editor and the authors to decide whether this is too much additional work, but I believe it is reasonable giving the coding capability gauged from the figures.

Thanks to Reviewer 1 for this good point. We fully agree, and have developed an improved strategy (Figure 4) for automated neighbour detection, that does not rely on cells being regular in shape/size. This new method works by slightly expanding the border of each cell in 3D by just ~10% of a cell diameter, such that now cell borders slightly overlap only with directly touching neighbour cells. The program then identifies all border intersections, producing a list of only directly touching neighbours. We verify that this new method achieves near perfect agreement with a manual neighbour count. All code for the new automated neighbour detection method is provided.

We also appreciate that for some analyses, comparing cell expression among a larger group of cells within a given area may be more meaningful than limiting only to touching neighbours. Accordingly, we have also provided supplementary code for calculating and filtering the Euclidian distance between cell centre coordinates, to identify all nearby cells within a user-specified search radius.

For changes, please see updated **Figure 4**, and lines **241-262**. All code is made available via https://github.com/LiliansCalvo/smiFISH_Arthropods.

Minor points:

1. When describing eve expression in Figure 1, do authors suggest "low" eve mRNA expressing cells have post-transcriptional regulation that further sharpen protein-level expression boundaries, or just to highlight the superb smFISH resolution comparing to immunofluorescence and traditional in situ?

It is just to highlight the resolution obtained with smiFISH. A great advantage of the technique is that a single mRNA spot is the same intensity and equally detectable irrespective of overall gene expression level.

This allows visualization of very low expressing cells, even just a few RNA per cell, such as those ‘low’ eve expressing cells between stripes. With traditional in-situ or immunofluorescence relying on measurement of overall fluorescence level, such cells would likely be classed as non-expressing, as their overall fluorescence level would not differ significantly from background.

2. When segmenting cells using Spectrin signal, how far was the sharp mid-section Spectrin signal was extrapolated? Any supporting evidence or argument to support the choice of the cell height?

For analysing cell to cell variability, it is essential that the RNA assignment to individual cells is accurate. Therefore, image stacks were taken to the basal limit of membrane ingression but not further; as it is not possible to assign any mRNAs located lower than this membrane limit to specific cells with certainty. In the image analysed in Figure 3, the first 20 slices of Spectrin staining was maintained, and slice 20 extrapolated for a further 28 slices to bottom of the stack defined as the basal limit of membrane ingression. This has now been made clearer in the text.

The choice of cell height will depend upon the specific question. If accuracy of cell to cell variability is critical, then Spectrin staining should not be extrapolated beyond the basal membrane ingression limit. However, if total mRNA number is more critical, with less focus on cell-to-cell variability, deeper stacks can be taken to encompass every mRNA including any below the basal membrane limit, and the sharp mid-section membrane extrapolated accordingly beyond the basal limit through the whole stack. In a new Supplementary Figure 2, we have included quantifications of this kind from an additional 12 blastoderm embryos, to assess biological variability of total *Kr* mRNA number using deeper stack depths beyond the membrane limit. Supplementary Figure 2g shows differing degrees of membrane ingression which was measured to sort embryos by age order. The degree of membrane extrapolation depends therefore on embryo age (degree of membrane ingression), and whether stack depth is intended to encompass just mRNAs within membranes, or also to include those below the membrane limit. These considerations have now been included in the main text.

For changes, please see lines **167-172, 221-231, 487-490**.

3. In Figure 1, posterior mRNA accumulation of the *nos* mRNA is marginal. Is this common?

Yes, the staining is consistent with previous studies; the important point lies in the timing of posterior *nos* accumulation during oogenesis. The egg chamber shown in Figure 1 is at stage 10, and previous studies show that the majority of posterior localization of *nos* mRNAs occurs during stages 13/14 of oogenesis, following nurse cell ‘dumping’ of mRNA (PMID: 12867026, PMID: 9895314, PMID: 7515724). We chose to show a stage 10 oocyte, because this demonstrates the sensitivity of smiFISH by capturing the earliest onset of posterior accumulation of *nos* mRNA, earlier than previously observed by classic in-situ (PMID: 7515724 and PMID: 9895314), and live imaging (stage 11 PMID: 12867026) but similar to previous FISH staining, (stage 10 PMID 25848747). The text referring to this figure has now been adjusted to highlight this point.

For changes, please see lines **114-123**.

4. The abbreviation MYA (million years ago) needs to be explained when first mentioned.

This has been fixed.

For changes, please see **Supplementary Figure 1**, and line **103**.

5. In Figure 2, the Hox gene multiplexing. Are bright foci representing active transcription bursts? Is it still possible to count mRNA dots in these foci? When plotting the distribution of mRNA expressions of these genes with direct evidence of active transcription, does the histogram have a long tail? Discussion of this point can be combined with Figure 3 discussion.

Hox genes are generally long, ranging from ~10kb (*Dfd*) to ~103kb (*Antp*), therefore tend to show large bright transcriptional sites, representing a localized accumulation of multiple nascent RNAs in the process of transcription along the gene length. Therefore these bright foci are not so much individual bursts as they are more persistent marks of ongoing active transcription through the gene length. Intensity of transcription sites

can in principle be quantified, compared to single mRNA spot intensity, and the nascent RNA number inferred (PMID: 25728770). This approach may be relevant for long genes where much RNA is held at the transcription site, and for questions concerning transcription dynamics. A note about transcription site quantification has now been included in the Figure 2 discussion.

For changes, please see lines 134-139.

Reviewer #2:

Calvo et al. have adapted an existing single molecule FISH method, previously developed for cultured cells, to whole mount preparations. They demonstrate that the method works for embryos and tissues from a handful of arthropods. Using deconvolution and spectral unmixing, they show that up to eight genes can be assessed in whole mounts. They present a membrane labeling protocol compatible with their FISH method. They also present two straightforward measurements of cell-to-cell variability.

The mRNA labeling method will certainly be regarded by the community as a valuable cost-saving measure. The compatibility with membrane labeling will also be very useful for cell segmentation. Demonstrating the application of spectral unmixing is also useful. These are valuable additions to the toolbox of single mRNA molecule detection. However, the quantification is not rigorous and detracts from the manuscript. The manuscript otherwise contains no novel conclusions. The work might be more appropriate for a methods-centric publication once the authors address concerns about the quantification and measurement error outlined below.

This work is indeed intended as a methods paper, and aims to present a methodology pipeline for multi-gene mRNA quantification and cell to cell variability analysis in whole embryos. We do not aim to draw any particular biological conclusions from the specific genes imaged and quantified.

1) To advance the use of single molecule FISH in whole mount tissues, it is essential to present the evidence that the procedure and subsequent analysis correctly measure mRNA density. But the mRNA counts per cell presented here are not consistent with previous studies. Further, to ascertain the extent of biological variability requires accounting for measurement error, but this has not been done. Sources of measurement error that the authors should account for include:

We agree that it is important to demonstrate accuracy of the quantitation method. The original smiFISH publication (PMID: 27599845) thoroughly assesses the accuracy of smiFISH detection in cell culture, so we did not originally consider it necessary to repeat validations of the smiFISH technique. However, since we are applying smiFISH to whole embryos, and using a different image analysis software (Imaris), we recognize that separate validations of quantification accuracy are important and will strengthen the work. We have therefore performed additional experimental work to assess the accuracy of our quantitation in different ways, and these validations are provided in a new Supplementary Figure 2.

While our numbers are lower than those reported in PMID: 23953111, our range of maxima are consistent with several other studies that quantify mRNAs of early patterning genes. Boettiger and Levine 2013 quantify *snail* mRNA/cell and report maxima ranging from ~150 to 300 per cell across multiple embryos at nc14 (PMID 23352665). Hoppe *et al.* 2020 quantify *u-shaped (ush)* and *hindsight (hnt)* mRNA/cell and report maxima ranging from ~80-280 (*ush*) and ~120-240 (*hnt*) per cell across multiple nc14 embryos (PMID 32758422). Bothma *et al.* 2014 report a maximum of ~250 *eve* mRNA/nucleus in *eve* stripe 2 at nc14 (PMID 24994903).

1 i) Detection failure. The authors should estimate the extent of under-counting using two sets of probes labeled with different colors and directed against the same mRNA. Moreover, at a glance the failure rate appears quite high in the case of *nos* mRNA, single mRNAs of which are densely distributed through the entire oocyte (PMID: 9895314, 25848747).

In Supplementary Figure 2 a & b, we present this suggested two colour test for *Kr*, and confirm strong agreement in mRNA/cell detected between the two probe sets, indicating minimal under-counting due to detection failure. This two colour test is a useful addition to the methodology pipeline we present, allowing

detection failure to be measured and factored into quantifications where absolute mRNA numbers are critical.

For changes, please see **Supplementary Figure 2** and lines **205-208**.

We do not agree that our failure rate for *nos* detection is high; instead we believe that it is an issue of developmental timing. We detect abundant *nos* mRNAs in the nurse cells, significant accumulation at the anterior oocyte edge, and marginal accumulation at the posterior pole, which is expected for a stage 10 egg chamber, and is consistent with previous studies. For example, one of the studies indicated (PMID: 9895314, Figure 7) and another (PMID: 7515724 Figure 2) use less sensitive classic in-situ, and failed to detect any posterior *nos* mRNA at stage 10, with first detection observed at stage 12, and the majority of accumulation at stages 13 and 14 (PMID: 7515724). Similarly, a live imaging study first detected marginal posterior *nos* mRNA accumulation at stage 11, again with the majority of accumulation occurring later at stages 12 and 13 (PMID: 12867026). The publication indicated (PMID: 9895314) states the following: “...*endogenous nos RNA are highly abundant in the nurse cells at stage 10 (asterisk). This RNA cannot be detected in the oocyte until after stage 10 when the nurse cells empty their contents into the oocyte (not shown).*” This publication does not show that *nos* mRNA is densely distributed throughout the **oocyte**; it shows that it is distributed throughout **embryos**. The other paper indicated (PMID 25848747) reports distribution of *nos* mRNAs throughout **embryos** (with accumulation at the posterior pole), but does not show dense distribution throughout the stage 10 **oocyte**; it shows increasing accumulation of *nos* RNAs at the posterior end of the oocyte, from stage 10 to stage 13. The paper states: ‘*Synthesized by the ovarian nurse cells, nos enters the oocyte en masse when the nurse cells “dump” their contents at the end of stage 10 and becomes distributed throughout the oocyte by diffusion and the concurrent streaming of the oocyte cytoplasm (ooplasm)*’. It also states ‘...we observed continuous accumulation of *nos* in granules at the posterior of the oocyte beginning at stage 10 of oogenesis. *nos*-containing granules increase in both number and mRNA content up until the oocyte reaches maturity at stage 14’. We therefore consider that our early stage 10 egg chamber staining is consistent with previous publications, and captures the earliest beginnings of posterior *nos* accumulation.

1 ii) mRNA crowding. mRNAs will be undercounted at a rate that increases with increasing density. This is not accounted for in the manuscript, but has a major impact on any conclusions about variability. The authors do appear to be undercounting the gap gene mRNAs. For example, *Kr* has a maximum density of around 1000 mRNA per cell, where the basal extent of the cell is taken at 12 microns below the cortex (PMID: 23953111).

We have directly addressed mRNA crowding experimentally in Supplementary Figure 2 c & d, by imaging *Kr* mRNAs with 40X objective then with 100X objective, through identical z depths in the same embryo, to determine whether at lower magnification spots are overcrowded and consequently undercounted. Visual inspection of Imaris spot detection shows successful separation of crowded touching spots at both magnifications. We found a slight increase in detection efficiency at 100X compared with 40X, (Supplementary Figure 2d, 40X mean=54, max=136, 100X mean=66, max=142, non-zero cells only). This difference is modest and cannot account for the difference between our numbers and the maximum 1000 mRNA per cell stated.

We further considered whether differences in either embryo age, the depth of imaging, or embryo to embryo biological variability, may account for the difference between our *Kr* mRNA numbers and the maximum ~1000/cell figure. Using identical settings, we quantified *Kr* mRNAs in an additional 12 blastoderm embryos, imaged beyond the basal limit of membrane ingression to z-depths of up to 24um, starting just above the most apical mRNAs and ending just below the most basal ones. This captured every mRNA through the cytoplasmic depth, in a variety of different aged embryos, as measured by the degree of membrane ingression. Across the 12 embryos, maximum mRNA/cell ranged from 123 to 337. Since deep imaging was used to ensure inclusion of every mRNA, it is clear that z-stack depth is not responsible for the difference between our *Kr* counts and the previously reported 1000 mRNA/cell maximum. The substantial biological variability we find between embryos imaged and quantified identically, is consistent with the wild-type embryo to embryo variability found for snail mRNAs in Boettiger and Levine 2013 (PMID 23352665). This biological variability may account for some of the difference up to ~200 mRNA/cell, but

still cannot reconcile the full difference with the 1000 mRNA/cell maximum figure.

For changes, please see **Supplementary Figure 2**, and lines **208-231, 487-490**.

1 iii) Tissue deformation upon processing and mounting. The fixing and mounting procedure can alter the apparent density of mRNAs and nuclei and can be a dominant source of measurement error (PMID: 23953111, Fig. S2). In the current manuscript, such deformation might well contribute to measurement error, given the clear spatial correlations in the variability shown in Fig. 5G. For example, Kr variability approaches the Poisson limit in the most rapidly expressing regions of the embryo, i.e. Fano factor = 1 (PMID: 23953111, Fig. S6); however, in the current manuscript, the Fano factor clearly exceeds 1 in this region, and the Fano factor exhibits spatial correlations around 4-5 nuclear diameters. This is consistent with local tissue deformations impacting the counts per cell.

We consider that the spatial correlations in variability arise from shared positional identity of the cell in the context of the expression pattern. Cells in similar regions have similar levels and combinations of upstream transcription factors present, and consequently would be expected to show similar behaviour in terms of expression variability. This seems evident from the fact that spatial patterns in variability closely match the gene expression patterns themselves. For this spatial correlation in variability to be explained by tissue deformation, the physical deformations would inexplicably need to occur in patterns matching the gene expression domains.

2) As a final point, the usefulness of the two new measurements of variability is not immediately clear. As a means of assessing variability between cells, it might be more informative to assign each cell a value based on how many standard deviations away from the "neighborhood mean count" that cell is found, and a p value to say whether that distance is significantly larger than expected. None of the measurements presented (including the Fano factor) help the reader understand whether the mRNA content of a given cell is significantly larger (or smaller) than expected.

We have tested this suggestion of expressing the variability as number of standard deviations the cell mRNA number is from the 'neighbourhood mean count', which is the Z-score. The attached variability measures spreadsheet below shows that the Z-score performs similarly to our numerical variability (NV), returning identical values for regions 1, 3 & 4, which have identical variability in absolute mRNA number, and a lower score for region 2. However, like NV, the Z-score does not differentiate the proportional difference between regions 3 and 4, whereas our proportional variability (PV) measure does reflect this difference. The Z-score does potentially have the advantage of assigning a p value to the score, however, this relies on the assumption that the data is normally distributed. For many cells, the neighbour number is either too low to test for normality, or the neighbour counts are non-normally distributed (for example, *giant*, normality test error: 36% of neighbour groups, non-normal: 10% of neighbour groups, normal: 53% of neighbour groups). Non-normality may be expected for genes expressed in discrete patterns, where some neighbour groups will lie on expression boundaries. The Z-score may be a useful alternative to NV, in applications where mRNA is being compared among larger groups of cells within a given area of tissue, as opposed to just touching neighbours, since a greater neighbour number would allow the assumption of a normal distribution to be tested. A normally distributed neighbour group may also be more likely for ubiquitously expressed genes, than genes expressed in discrete domains. The measures PV and NV do faithfully reflect absolute numerical variability and proportional variability of a given cell, with respect to its direct neighbours. To interpret the significance of these scores, statistics can subsequently be performed on the raw NV and PV values, depending the biological question. For example, cells with a NV or PV score significantly higher than the population mean score can be identified, to highlight the most 'abnormally variable' cells within the embryo. Additional text discussing this has now been included in the discussion section.

For changes, please see lines **348-355**.

Variability measures spreadsheet:

Region	Cell Count Data			Region Total	Region Mean	Region Variance	FF Variance/Mean	Neighbours	$\Sigma X1-Xn $	$\Sigma X1-Xn /n$	NV $(\Sigma X1-Xn /n) / \text{max-P}$	PV $(\Sigma X1-Xn /n) / \text{max-N}$	Neighbour Mean	Neighbour SD	Z-score Centre Cell SD from Neighbour Mean
1	100	101	101	808	89.78	1108.44	12.35	8	799	99.875	0.09	0.988861386	100.875	0.3535534	282.4891591
	101	1	101												
	101	101	101												
2	1	101	101	809	89.89	1111.11	12.36	8	100	12.5	0.01	0.123762376	88.5	35.355339	0.353553391
	101	101	101												
	101	101	101												
3	2	1	1	110	12.22	1108.44	90.69	8	799	99.875	0.09	0.988861386	1.125	0.3535534	282.4891591
	1	101	1												
	1	1	1												
4	1099	1100	1100	9799	1088.78	1108.44	1.02	8	799	99.875	0.09	0.090795455	1099.875	0.3535534	282.4891591
	1100	1000	1100												
	1100	1100	1100												

[revised manuscript text omitted]

outcomes.

**METHODS**

**Solutions**

50% bleach: 50% sodium hypochlorite solution in distilled H₂O. Embryo wash buffer: 0.1M NaCl,
0.02% triton X-100 in distilled H₂O. Fix solution: 0.5ml 10X PBS (Sigma, without CaCl₂ and MgCl₂),
0.5ml nuclease-free H₂O, 4ml ultrapure 10% methanol-free formaldehyde (Polysciences), 5ml
heptane 99% (Sigma). PBT: 1X PBS (Sigma, without CaCl₂ and MgCl₂), 0.05% tween-20 (Sigma), in
nuclease-free H₂O. smiFISH wash buffer: 2X SSC, 10% deionised formamide (Ambion) in nuclease-
free H₂O. smiFISH hybridization buffer: 10% w/v dextran sulphate (Sigma, molecular weight 6,500-
10,000), 2X SSC, 10% deionised formamide in nuclease-free H₂O. Blocking solution: 1X western
blocking reagent (Sigma) in PBT.

**Sample fixation**

*Drosophila* embryos collected on apple juice agar plates at 25°C were dechorionated with 50% bleach,
alternately washed with distilled water and embryo wash buffer, and shaken in fix solution at 240rpm
for 45 minutes. The aqueous solution layer was removed, 10ml 100% methanol added, and shaken for
1 minute to devitellinize embryos. Devitellinized embryos were washed 5x with 100% methanol, then
stored in 100% methanol at -20°C. Fixed *Tribolium* embryos were kindly provided by Olivia Tidswell
from Michael Akam's lab. *Tribolium* were dechorionated and fixed as described for *Drosophila*, but
for increased devitellinization efficiency, embryos were passed through a 19G needle in ice-cold
100% methanol. Fixed *Nasonia* embryos were kindly provided by Shannon Taylor from Peter
Dearden's lab¹⁸. *Parhyale* embryos were manually collected from females anaesthetized with 0.01%
clove oil in sea water. Embryos were washed 3x with filtered sea water, transferred to fix solution,
and shaken at 240rpm for 45 minutes. Embryos were then transferred to a glass dish in 1X PBS, for
manual removal of the chorion and vitelline membrane using tungsten needles. Dissected embryos
were transferred to a second fixation solution of 4% formaldehyde in 1X PBS for 1 hour, before
washing 5x in 100% methanol and storage in 100% methanol at -20°C. For imaginal discs, white pre-
pupae were chilled in 1°C 1X PBS, the cuticle opened longitudinally, and pupae fixed in a solution of
4% formaldehyde in 1X PBS for 1 hour (no rocking), before washing 5x in 100% methanol and
storage in 100% methanol at -20°C. For ovaries, adult females were thoroughly anaesthetized with

CO₂, placed in 1°C 1X PBT, and ovaries dissected out and opened to expose ovarioles. Dissected
ovaries were washed with 1X PBS, then fixed and stored as described above for pupae.

**Probe design and preparation**

*D.melanogaster* and *D.virilis* mRNA sequences were obtained from Flybase (<https://flybase.org>);
*T.castaneum*, *N.vitripennis* and *P.hawaiensis* mRNA sequences were obtained from NCBI
(<https://www.ncbi.nlm.nih.gov/nucleotide/>). Complementary 20nt DNA probes against mRNA
sequences (up to 48 probes per gene) were designed using the Biosearch Technologies stellaris RNA
FISH probe designer tool (free with registration, <https://biosearchtech.com>). All probe sequences are
provided in Supplementary Table 1. The following sequence was added to the 5' end of each 20nt
probe: CCTCCTAAGTTTCGAGCTGGACTCAGTG. This is the reverse complement of the X FLAP
sequence used in Tsanov *et al.* 2016. Oligos were ordered from Integrated DNA Technologies (IDT),
in 96 well plates, using 25nmole synthesis scale, standard desalting, and at 100µM in nuclease-free
H₂O. A standard volume (we use 50µl) of each probe was added together to generate an equimolar
probe mix, at 100µM (mixed probe concentration). A set of 48 probes therefore gave 48 x 50µl =
2.4ml of probe mix, which was split into 100µl aliquots for storage at -20°C. The X FLAP sequence
itself CACTGAGTCCAGCTCGAACTTAGGAGG was 5' and 3' end-labeled with CalFluor 540,
Quasar 570, CalFluor 590, CalFluor 610, CalFluor 635, Quasar 670, and Quasar 705, and synthesized
by Biosearch Technologies. X FLAP sequence 5' and 3' end-labeled with Alexa Fluor 488 was
synthesized by IDT. Lyophilized fluorophore-labelled FLAP was resuspended in nuclease-free H₂O to
a concentration of 100µM, aliquoted and stored at -20°C.

**Probe FLAP annealing**

Probes were annealed to fluorophore-labeled FLAP sequences in 50µl reactions, as follows:

Component	Volume per reaction	Amount
Probe set (100µM mixed probe concentration)	2µl	200 pmol (total DNA)
Fluorophore-labelled FLAP (100µM)	2.5µl	250 pmol
10X NEB 3*	5µl
nuclease-free H ₂ O.	40.5µl

*NEB 3: New England Biolabs Buffer 3 (1X composition: 100mM NaCl, 50mM Tris-HCl, 10mM
MgCl₂, 1 mM DTT, pH 7.9).

When more than 3 probe sets were to be used on the same sample, probes were annealed at 5x
concentration, so they could be used at 1/5 normal volume in the hybridization buffer, to avoid large

volumes of probe affecting salt and formamide concentration in the hybridization. The 5x
concentration annealing reactions were set up as follows:

Component	Volume per reaction	Amount
Probe set (100µM mixed probe concentration)	10µl	1 nmol (total DNA)
Fluorophore-labelled FLAP (100µM)	12.5µl	1.25 nmol
10X NEB 3*	5µl
nuclease-free H ₂ O.	22.5µl

Annealing reactions were performed in a thermal cycler according to Tsanov *et al.* 2016: Lid 99°C,
85°C 3 minutes, 65°C 3 minutes, 25°C 5 minutes, 4°C hold. Final annealed probes are at either 4µM
(from the normal concentration annealing reaction) or 20µM (from the 5x concentrated annealing
reaction). Annealed probes were stored at -20°C. Probe/fluorophore combinations are supplied in
Supplementary Table 2.

**smiFISH and immunofluorescence**

All steps were performed in glass Wheaton v-vials (Sigma). Using glass is important because embryos
and tissues stick to plastic in the smiFISH wash buffer and hybridization buffer, and the v-base helps
with sample visualization and retention through the multiple solution changes. Fixed samples stored
in 100% methanol were transitioned to PBT in stages: 50% PBT, 75% PBT, 100% PBT, 5 minutes
452 per wash. Samples were washed 3x 10 minutes in PBT, then 10 minutes in 50% PBT 50% smiFISH
wash buffer, before 2x 30 minute pre-hybridization washes in smiFISH wash buffer at 37°C.
Annealed smiFISH probes (4µM or 20µM) were diluted in 500µl smiFISH hybridization buffer to a
concentration of 80nM. Probes were hybridized with samples in the dark at 37°C for 14 hours.
Samples were washed 4x 15 minutes in smiFISH wash buffer at 37°C, then 3x 10 minutes in PBT at
room temperature. For immunofluorescence, samples were blocked for 30 minutes in blocking
solution, then incubated with anti *Drosophila* alpha-Spectrin (DSHB 3A9) diluted 1:50 in blocking
solution for 18 hours at 4°C. Samples were washed 4x 15 minutes with PBT, blocked for 30 minutes,
incubated with goat anti mouse Alexa Fluor 488 (ThermoFisher) diluted 1:500 in blocking solution
for 4 hours at room temperature, then washed 4x 15 minutes with PBT. In PBT, *Tribolium* and
*Parhyale* embryos were manually dissected away from yolk using tungsten needles, imaginal discs
were dissected away from pupal carcasses, and single ovarioles and egg chambers were dissected
away from one another. All samples were mounted under #1.5 coverslips using prolong diamond
antifade mountant with DAPI (ThermoFisher). Due to their size, coverslip spacers were required for
*Parhyale* embryos.

**Imaging**

Images were acquired on a Leica TCS SP8 AOBS inverted gSTED microscope using a 40x/1.3 or
100x/1.4 HC PL APO (oil) objective. Image stacks for each different species, imaginal discs and
ovaries were taken with the following settings: format 2048x2048 or 4096x4096, speed 400Hz
unidirectional, sequential line scanning, line averaging 8 or 16, pinhole 1 airy unit. Each channel was
gated 1.0-6.0. DAPI excitation 405nm, laser 5%, collection 415-480nm. CalFluor 610 excitation
590nm, laser 20%, collection 600-642nm. Quasar 670 excitation 647nm, laser 20%, collection 657-
750nm.

*D.melanogaster* embryo image stacks showing all 8 Hox genes (Figure 2), 5 segmentation genes
(Figure 3), or *Kr* mRNA for quantification validations (Supplementary Figure 2) were taken with the
following settings: format 4096x4096, speed 400Hz unidirectional, sequential line scanning, line
averaging 16, pinhole 1 airy unit. Each channel was gated 1.0-6.0. DAPI excitation 405nm, laser 5%,
collection 415-480nm. AlexaFluor 488 excitation 490nm, laser 15%, collection 498-530nm. CalFluor
540 excitation 522nm, laser 15%, collection 530-555nm. Quasar 570 excitation 548nm, laser power
15%, collection 558-575nm. CalFluor 590 excitation 569nm, laser 15%, collection 579-595nm.
CalFluor 610 excitation 590nm, laser 15%, collection 605-620nm. CalFluor 635 excitation 618nm,
laser 15%, collection 628-650nm. Quasar 670 excitation 647nm, laser 10%, collection 660-680nm.
Quasar 705 excitation 670nm, laser 10%, collection 695-780nm. Image stacks were acquired with a
200nm z interval. For images intended for mRNA quantification, z-stack limits were set to start just
above the most apical smiFISH signal, and to end either at the basal extent of membrane ingression
(to capture all RNAs that can accurately be assigned to single cells), or just below the basal extent of
smiFISH signal (to capture all mRNAs throughout the full cytoplasmic depth). Spectral unmixing of
the 8 Hox gene channels was performed in the Leica LAS X v1.8.0.13370 software, using a 30µm
radius selection in each channel to build the unmixing matrix.

**Image analysis**

Z stacks were stabilized through z to account for any imaging drift, and deconvolved using Huygens
Professional v18.04. For single-cell segmentation and mRNA quantification, DAPI, Spectrin, and
smiFISH image stacks were combined in Imaris v9.2 software. Spectrin staining forms a clear cell
border in z-slices where the cells are in cross-section, but fades out basally at the extent of membrane
ingression. The core set of z-slices that do show clear cross-sectional Spectrin staining was identified,
and the top and bottom slices of this core replicated to extend through the full depth of the stack,
replacing apical and basal slices with unclear cell borders. Cells were then segmented in 3D
automatically in the Imaris cells module from the Spectrin channel, using a smallest cell diameter of
5µm, membrane detail level of 0.5µm, and a local contrast filter. Edge cells and any double cells were

omitted by filtering the set of detected cells for outliers based on cell volume, sphericity and z-
position. smiFISH spots were detected using the spots function, allowing for different spot sizes, with
an estimated xy diameter of 0.3 μ m, estimated z diameter of 0.6 μ m, and background subtraction. Spot
quality thresholds were set individually for each channel, since brightness and diameter of spots is
inherently different between different fluorophores. Thresholds were set to just below the point at
which a sharp spike in background false positives outside of the established domain occurs. For
quantitation consistency, the same detection thresholds were used for embryo comparisons. To
prepare images for figures, maximum projections of smiFISH channels were generated in FIJI v 2.0.0-
rc-49/1.51d. Projections were then combined into RGB images in Adobe Photoshop CS6.

**STATISTICS AND REPRODUCIBILITY**

Statistics are not used within the main figures, but all cell n numbers are defined in the figure legends.
Details of the statistical analysis performed in Supplementary Figure 2, and all cell n numbers, are
provided in the legend. Heatmaps displaying mRNA/cell were generated in Imaris v9.2 software. Cell
variability heatmaps were generated in R using the package ggplot2. Statistical analyses and graphs
were completed using GraphPad Prism v 5.0.

**REPORTING SUMMARY**

Further information on research design is available in the Nature Research Reporting Summary linked
to this article.

**DATA AVAILABILITY**

All smiFISH probe sequences are available in Supplementary Table 1. All data underlying the graphs
and heatmaps presented can be accessed at [https://github.com/LliliansCalvo/smiFISH_Arthropods](https://github.com/LliliansCalvo/smiFISH_Arthropods).

**CODE AVAILABILITY**

All custom code for automated detection of neighbouring cells, together with tutorials, can be
accessed at: [https://github.com/LliliansCalvo/smiFISH_Arthropods](https://github.com/LliliansCalvo/smiFISH_Arthropods).

REFERENCES

- Gaspar, I. & Ephrussi, A. Strength in numbers: quantitative single-molecule RNA detection
assays. *Wiley Interdiscip Rev Dev Biol* **4**, 135-150, doi:10.1002/wdev.170 (2015).
- Adams, M. D. *et al.* Complementary DNA sequencing: expressed sequence tags and human
genome project. *Science* **252**, 1651-1656, doi:10.1126/science.2047873 (1991).
- Lister, R. *et al.* Highly integrated single-base resolution maps of the epigenome in
Arabidopsis. *Cell* **133**, 523-536 (2008).
- Mortazavi, A., Williams, B. A., McCue, K., Schaeffer, L. & Wold, B. Mapping and
quantifying mammalian transcriptomes by RNA-Seq. *Nature Methods* **5**, 621-628,
doi:10.1038/nmeth.1226 (2008).
- Schena, M., Shalon, D., Davis, R. W. & Brown, P. O. Quantitative monitoring of gene
expression patterns with a complementary DNA microarray. *Science* **270**, 467-470 (1995).
- Sandler, J. E. & Stathopoulos, A. Quantitative Single-Embryo Profile of Drosophila Genome
Activation and the Dorsal-Ventral Patterning Network. *Genetics* **202**, 1575-1584,
doi:10.1534/genetics.116.186783 (2016).
- Karaiskos, N. *et al.* The Drosophila embryo at single-cell transcriptome resolution. *Science*
**358**, 194-199, doi:10.1126/science.aan3235 (2017).
- Wang, N. *et al.* Single-cell microRNA-mRNA co-sequencing reveals non-genetic
heterogeneity and mechanisms of microRNA regulation. *Nat Commun* **10**, 95,
doi:10.1038/s41467-018-07981-6 (2019).
- Gerhardt, H. *et al.* VEGF guides angiogenic sprouting utilizing endothelial tip cell filopodia.
*The Journal of cell biology* **161**, 1163-1177 (2003).
- Ellis, S. J. *et al.* Distinct modes of cell competition shape mammalian tissue morphogenesis.
*Nature* **569**, 497-502, doi:10.1038/s41586-019-1199-y (2019).
- Rheume, B. A. *et al.* Single cell transcriptome profiling of retinal ganglion cells identifies
cellular subtypes. *Nature Communications* **9**, 2759, doi:10.1038/s41467-018-05134-3 (2018).
- Karaayvaz, M. *et al.* Unravelling subclonal heterogeneity and aggressive disease states in
TNBC through single-cell RNA-seq. *Nature Communications* **9**, 3588, doi:10.1038/s41467-
018-06052-0 (2018).
- Ziegenhain, C., Vieth, B., Parekh, S., Hellmann, I. & Enard, W. Quantitative single-cell
transcriptomics. *Brief Funct Genomics* **17**, 220-232, doi:10.1093/bfgp/ely009 (2018).
- Femino, A. M., Fay, F. S., Fogarty, K. & Singer, R. H. Visualization of single RNA
transcripts in situ. *Science* **280**, 585-590, doi:10.1126/science.280.5363.585 (1998).
- Orjalo, A. a. J., Hans E and Ruth, Jerry L. Stellaris™ fluorescence in situ hybridization (FISH)
probes: a powerful tool for mRNA detection *Nature Methods* **8**, i--ii, doi:10.1038/nmeth.f.349
(2011).
- Tsanov, N. *et al.* smiFISH and FISH-quant - a flexible single RNA detection approach with
super-resolution capability. *Nucleic Acids Res* **44**, e165, doi:10.1093/nar/gkw784 (2016).
- Jones, D. L., Brewster, R. C. & Phillips, R. Promoter architecture dictates cell-to-cell
variability in gene expression. *Science* **346**, 1533-1536, doi:10.1126/science.1255301 (2014).
- Taylor, S. E. *et al.* The torso-like gene functions to maintain the structure of the vitelline
membrane in *Nasonia vitripennis*, implying its co-option into *Drosophila* axis formation. *Biol*
*Open* **8**, doi:10.1242/bio.046284 (2019).
- Frasch, M., Hoey, T., Rushlow, C., Doyle, H. & Levine, M. Characterization and localization
of the even-skipped protein of *Drosophila*. *EMBO J* **6**, 749-759 (1987).
- Lim, B., Fukaya, T., Heist, T. & Levine, M. Temporal dynamics of pair-rule stripes in living
*Drosophila* embryos. *Proc Natl Acad Sci U S A* **115**, 8376-8381,
doi:10.1073/pnas.1810430115 (2018).
- Fujioka, M., Jaynes, J. B. & Goto, T. Early even-skipped stripes act as morphogenetic
gradients at the single cell level to establish engrailed expression. *Development* **121**, 4371-
4382 (1995).
- Patel, N. H., Condrón, B. G. & Zinn, K. Pair-rule expression patterns of even-skipped are
found in both short- and long-germ beetles. *Nature* **367**, 429-434, doi:10.1038/367429a0
(1994).

- Rosenberg, M. I., Brent, A. E., Payre, F. & Desplan, C. Dual mode of embryonic
development is highlighted by expression and function of *Nasonia* pair-rule genes. *Elife* **3**,
e01440, doi:10.7554/eLife.01440 (2014).
- Vargas-Vila, M. A., Hannibal, R. L., Parchem, R. J., Liu, P. Z. & Patel, N. H. A prominent
requirement for single-minded and the ventral midline in patterning the dorsoventral axis of
the crustacean *Parhyale hawaiiensis*. *Development* **137**, 3469-3476, doi:10.1242/dev.055160
(2010).
- Pokrywka, N. J. & Stephenson, E. C. Microtubules mediate the localization of bicoid RNA
during *Drosophila* oogenesis. *Development* **113**, 55-66 (1991).
- Cha, B. J., Koppetsch, B. S. & Theurkauf, W. E. In vivo analysis of *Drosophila* bicoid mRNA
localization reveals a novel microtubule-dependent axis specification pathway. *Cell* **106**, 35-
46, doi:10.1016/s0092-8674(01)00419-6 (2001).
- Wang, C., Dickinson, L. K. & Lehmann, R. Genetics of nanos localization in *Drosophila*. *Dev*
*Dyn* **199**, 103-115, doi:10.1002/aja.1001990204 (1994).
- Dahanukar, A. & Wharton, R. P. The Nanos gradient in *Drosophila* embryos is generated by
translational regulation. *Genes Dev* **10**, 2610-2620, doi:10.1101/gad.10.20.2610 (1996).
- Bergsten, S. E. & Gavis, E. R. Role for mRNA localization in translational activation but not
spatial restriction of nanos RNA. *Development* **126**, 659-669 (1999).
- Forrest, K. M. & Gavis, E. R. Live imaging of endogenous RNA reveals a diffusion and
entrapment mechanism for nanos mRNA localization in *Drosophila*. *Curr Biol* **13**, 1159-1168,
doi:10.1016/s0960-9822(03)00451-2 (2003).
- Little, S. C., Sinsimer, K. S., Lee, J. J., Wieschaus, E. F. & Gavis, E. R. Independent and
coordinate trafficking of single *Drosophila* germ plasm mRNAs. *Nat Cell Biol* **17**, 558-568,
doi:10.1038/ncb3143 (2015).
- Bahar Halpern, K. *et al.* Bursty gene expression in the intact mammalian liver. *Mol Cell* **58**,
147-156, doi:10.1016/j.molcel.2015.01.027 (2015).
- Munsky, B., Neuert, G. & van Oudenaarden, A. Using gene expression noise to understand
gene regulation. *Science* **336**, 183-187, doi:10.1126/science.1216379 (2012).
- Warren, L. *et al.* Highly efficient reprogramming to pluripotency and directed differentiation
of human cells with synthetic modified mRNA. *Cell stem cell* **7**, 618-630 (2010).
- Zhong, X. *et al.* Circadian Clock Regulation of Hepatic Lipid Metabolism by Modulation of
m(6)A mRNA Methylation. *Cell Rep* **25**, 1816-1828 e1814, doi:10.1016/j.celrep.2018.10.068
(2018).
- Levine, E., Zhang, Z., Kuhlman, T. & Hwa, T. Quantitative characteristics of gene regulation
by small RNA. *PLoS biology* **5** (2007).
- Phillips, N. E. *et al.* Stochasticity in the miR-9/Hes1 oscillatory network can account for
clonal heterogeneity in the timing of differentiation. *Elife* **5**, doi:10.7554/eLife.16118 (2016).
- Jafar-Nejad, H. *et al.* Senseless acts as a binary switch during sensory organ precursor
selection. *Genes Dev* **17**, 2966-2978, doi:10.1101/gad.1122403 (2003).
- Papadopoulos, D. K. *et al.* Control of Hox transcription factor concentration and cell-to-cell
variability by an auto-regulatory switch. *Development* **146**, doi:10.1242/dev.168179 (2019).

**ACKNOWLEDGEMENTS**

[revised manuscript text omitted]

Variability measure	Formula	Definition of terms
Fano factor (FF)	$\frac{\text{Variance}_n}{\text{mean}_n}$	Variance_n - Variance of the immediate neighbour group, including centre cell mean_n - mean of the immediate neighbour group, including centre cell
Local numerical cell variability (NV)	$\frac{\sum (x_1 - x_n)/n}{\max_p}$	x_1 - mRNA number of the centre cell x_n - mRNA number of each immediately neighbouring cell n - number of immediate neighbours, NOT including centre cell
Local proportional cell variability (PV)	$\frac{\sum (x_1 - x_n)/n}{\max_n}$	\max_p - maximum mRNA number of all cells in the population \max_n - maximum mRNA number of immediate neighbours, including centre cell

Figure 5. Three alternative measures to express cell to cell mRNA variability.

Figure 5. Three alternative measures to express cell to cell mRNA variability. a) Formulae for Fano factor, a commonly used measure of cell variability, and NV and PV, two alternative measures designed to better capture individual cell variability. b) Cells from the anterior stripe of *eve* expression in *P.hawaiensis* early germband embryo, highlighting that individual cells can differ greatly in expression from their immediate neighbours. The centre cell (white arrow) in the left panel is similar to all its neighbouring cells except one, whereas the centre cell in the right panel is highly different from all of its neighbours except one. c-f) Hypothetical scenarios of neighbour group mRNA variability, to highlight the capacity of each formula to capture the variability of the single centre cell. 550 mRNA/cell is used as the population maximum. Fano factor incorrectly returns the same value for c and d, and incorrectly finds e to be more variable than c. NV correctly returns the same value for c, e and f, and a low value for d. PV correctly returns the same value for c and e, and lower values for d and f. g) Heatmaps show the three different variability measures, calculated for each segmented cell of the embryo (1982 cells in total), for five genes, *even-skipped*, *hunchback*, *knirps*, *giant* and *Kruppel*. Dots representing cells are scaled in both size and colour by the variability value. Both NV and PV measures can range from minimum 0 to maximum 1. For PV heatmaps, cells were filtered on the criteria of neighbour group mean mRNA number ≥ 1 ; cells failing this criterion were assigned a score of 0.

Supplementary Figure 1. Evolutionary divergence times of different arthropod model species.
 The species used in this study are highlighted in red, and belong to the clade pancrustacea, which emerged ~530 million years ago (MYA), and comprises all hexapods and crustaceans.

Supplementary Figure 2. Accuracy validations of mRNA quantification.

Supplementary Figure 2. Accuracy validations of mRNA quantification. a & b) Two colour detection efficiency test. **a)** smiFISH using two interleaved probe sets (each 41 probes) against *Kr* mRNA, labelled in Quasar 570 and CalFluor 610. Spots were imaged using 100X objective, through 48 slices with z step size of 200nm. **b)** Spots in each channel were detected in Imaris, and assigned to the 637 segmented cells. Correlation in *Kr* mRNA/cell detected with each probe set was measured (two tailed Spearman ranked correlation coefficient $r = 0.99$, $P < 0.0001$). **c & d)** Two magnification detection efficiency test. **c)** smiFISH of *Kr* mRNAs using CalFluor 610. The same stage 5 blastoderm embryo was imaged with 40X objective and then 100X objective, through the same z depth of 48 slices with z step size of 200nm. Spot detection in Imaris using identical settings for both magnifications detected both strong and weak spots, successfully separated closely touching spots, with minimal false positives as indicated by the minimal spot detection towards the edge of the *Kr* stripe. **d)** *Kr* mRNA/cell at each magnification, non-zero cells only, each dot in the plot represents a cell, horizontal lines are the mean, error bars show the 95% confidence interval of the mean. There is a slight increase in detection efficiency at 100X compared with 40X (40X mean=54, max=136, 100X mean=66, max=142), showing a small degree of undercounting at 40X likely due to mRNA overcrowding. **e-h)** Biological variability in total *Kr* mRNA number. **e)** smiFISH of *Kr* mRNAs using CalFluor 610, imaged with 40X objective, beyond the basal limit of membrane ingression through 120 slices with z step size of 200nm, to capture every *Kr* mRNA in the cytoplasmic depth. **f)** Slice by slice profile of sum greyscale intensity across a region of interest (ROI) through 120 z-slices. The trough between slices ~40-60 corresponds to the nucleus. **g)** Spectrin membrane staining at a cross sectional plane shows differing degrees of membrane ingression between stage 5 blastoderm embryos, used as a measure of embryo age. **h)** smiFISH of *Kr* mRNAs using CalFluor 610, in 12 blastoderm embryos imaged with 40X objective, z step size of 200nm, through total z-depths ranging from 20 μ m to 24 μ m, set to extend from above the apical extent to below the basal extent of *Kr* mRNA spots. For cell segmentation, Spectrin membrane staining was extrapolated beyond the basal ingression limit, to the full stack depth. Identical Imaris spot detection settings were used across all embryos. The plot shows *Kr* mRNA/cell across the 12 blastoderm embryos arranged in ascending age order, as measured by the degree of membrane ingression. Each dot in the plot represents a cell, horizontal lines are the mean, error bars show the 95% confidence interval of the mean. Cell numbers: Embryo 1: 1526, 2: 1686, 3:1827, 4: 1923, 5: 1248, 6: 1797, 7: 1897, 8: 1834, 9: 1717, 10: 1641, 11: 1662, 12: 1830.

Figure	Species	Sample type	Gene	Fluorophore
1	Drosophila melanogaster	embryo	even-skipped (eve)	CalFluor 610
1	Drosophila melanogaster	embryo	engrailed (en)	Quasar 670
1	Drosophila virilis	embryo	even-skipped (eve)	CalFluor 610
1	Drosophila virilis	embryo	engrailed (en)	Quasar 670
1	Tribolium castaneum	embryo	even-skipped (eve)	CalFluor 610
1	Tribolium castaneum	embryo	engrailed (en)	Quasar 670
1	Nasonia vitripennis	embryo	even-skipped (eve)	CalFluor 610
1	Parhyale hawaiiensis	embryo	even-skipped (eve)	CalFluor 610
1	Parhyale hawaiiensis	embryo	engrailed (en)	CalFluor 610
1	Drosophila melanogaster	imaginal disc	wingless (wg)	CalFluor 610
1	Drosophila melanogaster	imaginal disc	engrailed (en)	Quasar 670
1	Drosophila melanogaster	ovary	bicoid (bcd)	CalFluor 610
1	Drosophila melanogaster	ovary	nanos (nos)	Quasar 670
2	Drosophila melanogaster	embryo	labial (lab)	CalFluor 610
			proboscipedia (pb)	Quasar 570
			Deformed (Dfd)	AlexaFluor 488
			Sex combs reduced (Scr)	Quasar 670
			Antennapedia promoter 1 (Antp P1)	CalFluor 540
			Ultrabithorax (Ubx)	Quasar 705
			abdominal-A (abd-A)	CalFluor 590
			Abdominal-B (Abd-B)	CalFluor 635
3 and 5	Drosophila melanogaster	embryo	even-skipped (eve)	Quasar 705
			hunchback (hb)	CalFluor 610
			knirps (kni)	CalFluor 590
			giant (gt)	Quasar 670
			Kruppel (Kr)	Quasar 570
Supp. 2	Drosophila melanogaster	embryo	Kruppel (Kr) interleaved set 1	CalFluor 610
			Kruppel (Kr) interleaved set 2	Quasar 570

Supplementary Table 2. All probe-fluorophore combinations used in this study.

REVIEWERS' COMMENTS:

Reviewer #1 (Remarks to the Author):

The revised manuscript has adequately addressed all my previously raised points. In particular, the new neighbor-finding algorithm relying on the R geometry package “sf” is quite elegant and shown by the authors to agree very well with manual detection. I could reproduce the graphs and results using the code and data provided on Github, but the clarity of tutorial can be improved. I recommend its publication while suggesting the following minor points to be fixed:

1. Tutorial of the neighbor-finding R code is not clear. Here are some suggestions:

(a) The R scripts, instead of a text dump of the console output, should be shared.

(b) Required packages should be listed at the top. Currently, several required packages were not made clear in the polygon method of neighbor finding, including magrittr, dplyr, tidyr, and data.table.

(c) More code comments should be added to explain the data structure and operations to guide non-experienced users. For example, the starting data frame of (x,y,z) positions of Spectrin spots were obtained from Imaris by segmenting Spectrin staining, which should be made clear ideally in a comment when reading in the csv file. What is the data structure returned by the convex hull function? Does the `st_convex_hull` operate in 3D? Or does it only consider the (x,y) so effectively operate on a 2D projection?

(d) I suggest using relative path so that others can reproduce your R commands easily after cloning the git repo. For example, if R scripts are in a “scripts” folder side by side with the “Raw_data” folder, use something like `setwd(dirname(rstudioapi::getActiveDocumentContext()$path))` to set the working directory to the script folder, then use `read.csv('../Raw_data/the_csv_file.csv')`.

(e) Other codes to reproduce heatmaps and other plots should also be shared. For example, the codes calculating and plotting cell variability scores.

2. Scale bars are missing in all figures. Please add scale bars and specify the bar length in microns in figure legends.

3. When describing extrapolation of Spectrin segmentation, please use microns instead of number of slices so that readers don't have to figure out you used 200 nm z-interval.

4. To be picky of wording, white light illumination is not laser, and should not be described as “white light laser.”

Reviewer #2 (Remarks to the Author):

The revised manuscript is appropriate for publication.

Reviewer #1 (Remarks to the Author):

The revised manuscript has adequately addressed all my previously raised points. In particular, the new neighbor-finding algorithm relying on the R geometry package “sf” is quite elegant and shown by the authors to agree very well with manual detection. I could reproduce the graphs and results using the code and data provided on Github, but the clarity of tutorial can be improved. I recommend its publication while suggesting the following minor points to be fixed:

1. Tutorial of the neighbor-finding R code is not clear. Here are some suggestions:

(a) The R scripts, instead of a text dump of the console output, should be shared.

This has been fixed.

(b) Required packages should be listed at the top. Currently, several required packages were not made clear in the polygon method of neighbor finding, including magrittr, dplyr, tidy, and data.table.

This has been fixed.

(c) More code comments should be added to explain the data structure and operations to guide non-experienced users. For example, the starting data frame of (x,y,z) positions of Spectrin spots were obtained from Imaris by segmenting Spectrin staining, which should be made clear ideally in a comment when reading in the csv file. What is the data structure returned by the convex hull function? Does the st_convex_hull operate in 3D? Or does it only consider the (x,y) so effectively operate on a 2D projection?

This has been fixed. More comments were added to explain each operation of the code, code was added to show the data structure, and we created a small test 3D data set to confirm that the neighbour finding code operates in 3D.

(d) I suggest using relative path so that others can reproduce your R commands easily after cloning the git repo. For example, if R scripts are in a “scripts” folder side by side with the “Raw_data” folder, use something like “setwd(dirname(rstudioapi::getActiveDocumentContext()\$path))” to set the working directory to the script folder, then use “read.csv('./Raw_data/the_csv_file.csv').”

Thanks for the great suggestion, this has been implemented.

(e) Other codes to reproduce heatmaps and other plots should also be shared. For example, the codes calculating and plotting cell variability scores.

This has now been done. Code to reproduce heatmaps has been added. Variability scores were calculated in excel, and the excel files containing these calculation formulae have been provided.

2. Scale bars are missing in all figures. Please add scale bars and specify the bar length in microns in figure legends.

Scale bars in microns have now been added to all microscopy images.

3. When describing extrapolation of Spectrin segmentation, please use microns instead of number of slices so that readers don't have to figure out you used 200 nm z-interval.

Microns have now been included throughout for any specifications of z-depths.

4. To be picky of wording, white light illumination is not laser, and should not be described as “white light laser.”

Leica themselves use the terminology ‘white light laser’ when describing this type of illumination: <https://www.leica-microsystems.com/science-lab/white-light-laser/>, and it is accurate terminology because it works by a laser initially producing monochromatic light, which is subsequently spread into a broad spectrum. Therefore I did not see it as accurate to change this, so have kept “white light laser” in the text.

Reviewer #2 (Remarks to the Author):

The revised manuscript is appropriate for publication.